

# Seasonality in the Δ³³S measured in urban aerosols highlights an additional oxidation pathway for atmospheric SO₂

David Au Yang[1,2], Pierre Cartigny[1], Karine Desboeufs[3], David Widory[2]

[1] Laboratoire de Géochimie des Isotopes Stables, Institut de Physique du Globe de Paris, Université Paris Diderot, CNRS UMR 7154, Sorbonne Paris-Cité, 1 rue de Jussieu, 75005 Paris, France
[2] GEOTOP/ Université du Québec à Montreal, Montreal H3C 3P8, Canada
[3] Laboratoire Interuniversitaire des Systèmes Atmosphériques (LISA), UMR7583 CNRS, Université Paris 7 Denis Diderot, Université Paris-Est Créteil, Institut Pierre-Simon Laplace, Créteil, 94010, France

*Correspondence to*: David Au Yang (auyang@mail.gyig.ac.cn)

**Abstract.** Sulfates present in urban aerosols collected worldwide usually exhibit significant non-zero Δ³³S signatures (from -0.6 to 0.5‰) whose origin still remains unclear. To better address this issue, we recorded the seasonal variations of the multiple sulfur isotope compositions of PM₁₀ aerosols collected over the year 2013 at five stations within the Montreal Island (Canada), each characterized by distinct types and levels of pollution. The δ³⁴S-values (n=155) vary from 2.0 to 11.3‰ (± 0.2‰, 2σ), the Δ³³S-values from -0.080 to 0.341‰ (± 0.01‰, 2σ) and the Δ³⁶S-values from -1.082 to 1.751‰ (± 0.2‰, 2σ). Our study evidences a seasonality for both the δ³⁴S and Δ³³S, which can be observed either when considering all monitoring stations or, to a lesser degree, when considering them individually. Among them, the monitoring station located at the most western end of the island, upstream of local emissions, yields the lowest mean δ³⁴S coupled to the highest mean Δ³³S-values. The Δ³³S-values are higher during both summer and winter, and are < 0.1‰ during both spring and autumn. As these higher Δ³³S-values are measured in "upstream" aerosols, we conclude that the mechanism responsible for these highly positive S-MIF also occurs outside and not within the city, at odds with common assumptions. While the origin of such variability in the Δ³³S-values of urban aerosols (i.e. -0.6 to 0.5‰) is still subject to debate, we suggest that oxidation by Criegee radicals and/or photooxidation of atmospheric SO₂ in presence of mineral dust may play a role in generating such large ranges of S-MIF.

## 1 Introduction

Sulfur (S) is an element of environmental interest due to its key role in climate change and air pollution (Seinfeld and Pandis, 2012). Indeed, gaseous sulfur-bearing compounds are ubiquitous in both the marine and urban environments (Bardouki et al., 2003;Wall et al., 1988). The SO₂ which could come from direct emissions or which result from the oxidation of H₂S, dimethylsulfide (DMS) is the most common S-bearing gas in the atmosphere of which about 60% is deposited and eliminated from the atmosphere (Berglen et al., 2004;Harris et al., 2013b). The remaining 40% are ultimately oxidized into sulfate (SO₄²⁻) following two major chemical pathways (gaseous and aqueous) that will actually condensate, forming secondary sulfate aerosols (Seinfeld and Pandis, 2012;Tomasi and Lupi, 2017). The sulfate particles have chemical compositions ranging from





sulfuric acid droplets to ammonium sulfates (Sinha et al., 2008), depending on the availability of gaseous ammonia to neutralize the sulfuric acid. Sulfate aerosols affect both human health and climate (Albrecht, 1989;Lelieveld et al., 2015;Levy et al., 2013;Myhre et al., 2013;Penner et al., 1992;Penner et al., 2006;Ramanathan et al., 2005;Ramanathan et al., 2001). The chemical pathway converting precursors to sulfates is important because it changes radiative effects in the atmosphere. The

gaseous phase oxidation, which occurs predominantly via OH, permits the formation of new sulfate particles by homogeneous nucleation process (Benson, 2008;Kulmala et al., 2004;Tanaka et al., 1994). This nucleation via gas-to-particle conversion is the largest source of atmospheric aerosol particles in the free troposphere (Kulmala et al., 2004), providing up to half of the global cloud condensation nuclei (Merikanto et al., 2009;Yu and Luo, 2009). The $SO_2$ aqueous phase oxidation which occurs via several oxidants, the major ones being $O_2$+TMI (Transition Metal Ion), $H_2O_2$, $O_3$ and $NO_2$ (Alexander et al., 2012;Alexander

et al., 2009;Cheng et al., 2016;Harris et al., 2013a;Harris et al., 2013b;Herrmann, 2003;Sarwar et al., 2013;Seinfeld and Pandis, 2012;Lee and Schwartz, 1983), produces sulfates which will be released during the evaporation of cloud water. These sulfates will condense on pre-existing particles present in the cloud droplets (Mertes et al., 2005a;Mertes et al., 2005b). By causing some to condense onto larger particles with lower scattering efficiencies and shorter atmospheric lifetimes, the heterogeneous reactions of $SO_2$ on mineral aerosols (Andreae and Crutzen, 1997) and the oxidation of $SO_2$ into sulfate in sea-salt-containing

cloud droplets and deliquesced sea-salt aerosols reduce the radiative impact of sulfate aerosols.

Sulfur has four stable isotopes, $^{32}S$, $^{33}S$, $^{34}S$ and $^{36}S$ whose natural abundances are approximately 95%, 0.75%, 4.2% and 0.015%, respectively (Ding et al., 2001). The S-isotope compositions are expressed using the δ-notation defined as (Coplen, 2011):

$$\delta^{3x}S = \left( \frac{\left( ^{3x}S / _{32}S \right)_{sample}}{\left( ^{3x}S / _{32}S \right)_{CDT}} \right) - 1 ,$$

(1)

where $^{3x}S$ is one of the S heavy isotopes ($^{33}S$, $^{34}S$ or $^{36}S$) and CDT is the Vienna Canyon Diablo Troilite $^{34}S/^{32}S$ international standard. There is no international standard for the $^{33}S/^{32}S$ and $^{36}S/^{32}S$. Accuracy of the measured values is established by direct comparison with data measured by other laboratories.

Stable isotopes fractionate during unidirectional (kinetic) and/or exchange (equilibrium) reactions, resulting in the product and

reactant having distinct isotope compositions. Isotope fractionation factors are expressed using the α-notation, which relates the two isotope compositions as follows (expressed here for the $SO_2$ oxidation into $SO_4$) (Coplen, 2011):

$$^{3x}\alpha_{sulfate-SO_2} = \frac{\left( ^{3x}S / _{32}S \right)_{sulfate}}{\left( ^{3x}S / _{32}S \right)_{SO_2}} = \frac{\delta^{3x}S_{sulfate}+1}{\delta^{3x}S_{SO_2}+1}$$

(2)

At equilibrium, the three sulfur isotope ratios are usually scaled to each other according to their mass $((1/m_1-1/m_2)/(1/m_1-1/m_3))$, following a "mass-dependent fractionation" model (Farquhar et al., 2000). The isotope fractionation of $^{33}S$ over $^{32}S$ (1





amu difference) has approximately half the magnitude of the fractionation of the $^{34}S$ over $^{32}S$ (2 amu difference). More rigorously, mass-dependent fractionation is expressed by (Young et al., 2002;Dauphas and Schauble, 2016):

$$^{3y}\alpha = \left(^{34}\alpha\right)^{3y\beta} \tag{3}$$

where $^{3y}\alpha$ is either $^{33}\alpha$ or $^{36}\alpha$ and $^{3y}\beta$ is either $^{33}\beta$ or $^{36}\beta$. The $^{3y}\beta$-exponent describes the relative fractionation of $^{3y}S/^{32}S$ and $^{34}S/^{32}S$. This value depends on the reaction considered (Farquhar et al., 2001;Harris et al., 2013a;Ono et al., 2013;Watanabe et al., 2009). At high temperature (> 500°C), $^{33}\beta$ and $^{36}\beta$-values are respectively 0.515 and 1.889 (Eldridge et al., 2016;Otake et al., 2008). Deviation of the $^{3y}\beta$-value from these high temperature values usually leads to non-zero $\Delta^{33}S$ and $\Delta^{36}S$ values typically in the range of ±0.1‰ and ±1‰, respectively. $\Delta^{33}S$ and $\Delta^{36}S$ are expressed as follows (Farquhar and Wing (2003):

$$\Delta^{33}S = \left(\delta^{33}S + 1\right) - \left(\delta^{34}S + 1\right)^{0.515} \tag{4}$$

$$\Delta^{36}S = \left(\delta^{36}S + 1\right) - \left(\delta^{34}S + 1\right)^{1.889} \tag{5}$$

Non-zero $\Delta^{33}S$-values can also result from non-equilibrium (kinetic) processes and their combination through Rayleigh fractionation and other mass conservation effects (Farquhar et al., 2007;Ono et al., 2006a;Harris et al., 2013a).

Previous studies showed that sulfates in urban aerosols display $\delta^{34}S$-values from 0 to 20‰ and $\Delta^{33}S$-values from -0.6 to 0.5‰ (Guo et al., 2010;Han et al., 2017;Romero and Thiemens, 2003;Shaheen et al., 2014;Lin et al., 2018b). Two main lines of reasoning are usually evoked to explain such S-isotope compositions. The first one neglects the role of S-isotope fractionation and uses S-isotopes as a direct tracer of emission sources that have been shown to be characterized by large and distinct $\delta^{34}S$-values (Becker and Hirner, 1998;Calhoun et al., 1991;Gaffney et al., 1980;Guo et al., 2016;Newman and Forrest, 1991;Nielsen, 1974;Norman et al., 2006;Premuzic et al., 1986;Smith and Batts, 1974;Wadleigh et al., 1996;Wasiuta et al., 2015). For example, sea-salt sulfate is characterized by a $\delta^{34}S$-value of 21‰ (Rees et al., 1978), marine biogenic non-sea salt sulfate has a $\delta^{34}S$ ranging from 12 to 19‰ (Calhoun et al., 1991;Sanusi et al., 2006;Oduro et al., 2012), while anthropogenic sulfur emissions are often lighter although there are significant variations between sources ranging from -40 to 30‰ (Nielsen, 1974;Norman et al., 2006;Wasiuta et al., 2015;Krouse and Grinenko, 1991). The alternative interpretation relies on a constant $SO_2$ isotope composition. The variations observed in the sulfur multiple isotope compositions ($\delta^{34}S$ and $\Delta^{33}S$) of rural sulfate aerosols reflect changes in the atmospheric concentrations of $SO_2$ oxidants, each having distinct fractionation factors (Harris et al., 2012b;Harris et al., 2012d;Harris et al., 2013a;Harris et al., 2013b). In this case, high (up to ~7‰) and low (down to ~1‰) $\delta^{34}S$-values are predicted during winter and summer, respectively (Harris et al., 2013a).

However to date, these -0.6 to 0.5‰ $\Delta^{33}S$-values reported in urban aerosols cannot be fully explained by a source effect, given that corresponding isotope compositions for emission sources vary from -0.2 to 0.2‰ (Lee et al., 2002). These also cannot be explained by the experimentally determined $^{34}\alpha$ and $^{33}\alpha$-values currently available in the literature (implicating $O_2$+TMI, $H_2O_2$ and OH; Harris et al. (2013a) and $NO_2$ Au Yang et al. (2018)) or their potential combination that predict $\Delta^{33}S$-values centered around 0‰ ; i.e. at odds with available data for urban aerosols (Guo et al., 2010;Han et al., 2017;Romero and Thiemens, 2003;Shaheen et al., 2014;Lin et al., 2018b).



Non-zero $\Delta^{33}$S-values may thus ultimately be attributed to urban-specific chemical reactions linked to the polluted urban environment. With that in mind and using Montreal as our study site, our objectives were i) to identify where the most positive $\Delta^{33}$S-values are produced: outside or inside the city, and ii) to characterize the $\Delta^{33}$S urban seasonality and decipher whether local emissions tend to increase/decrease the $\Delta^{33}$S-values.

## 2    Material & Methods

### 2.1    Sampling site

$PM_{10}$ aerosols (particles with an aerodynamical diameter <10 µm) were sampled over a one-year period in 2013 by the RSQA (Réseau de Surveillance de la Qualité de l'Air) in the city of Montreal (Canada; 45°N 73°W) and its vicinity. Montreal is considered as a relatively lowly-polluted city with an average annual $PM_{10}$ concentration of 16 µg.m$^{-3}$ (World Health

Organization, 2016). Montreal therefore respects the 20 µg.m$^{-3}$ guidelines set by the World Health Organization (WHO), while also exhibiting local variations with several stations recording concentrations punctually exceeding the mean 50 µg.m$^{-3}$ 24-hour guidelines (Boulet and Melançon, 2012, 2013).

Five monitoring stations (03, 06,13, 50, and 98) disseminated onto the Montreal island were selected: i) Station 03 "Saint-Jean-Baptiste" is located at the North-East end of the island and due to the dominant wind directions is likely more influenced

by local petro-chemistry industries. ii) Station 06 "Anjou" is located close to a two high-traffic highways interchange (highways 40 and 25). iii) Downtown station 13 "Drummond" represents the urban background. iv) Station 50 "Hochelaga-Maisonneuve", located at the Old-Port of Montreal, is expected to be influenced by maritime activities. v) Station 98 "Sainte-Anne de Bellevue" is at the western end of the island in a semi-rural environment about 35 km upstream of Montreal downtown and is thus less likely impacted by the global city anthropogenic emissions (Boulet and Melançon, 2012). The four stations 03,

06, 13 and 50 are thus usually considered as representative of the different polluted environments in the city while station 98 is more akin to sample aerosols generated elsewhere upstream and transported to Montreal.

$PM_{10}$ samples were weekly collected on pre-combusted quartz filters using a high volume $PM_{10}$ size selective inlet, with an average flow of 1.13 m$^3$.min$^{-1}$ for a period of 24 h, i.e. a sampled air volume around 1627 m$^3$ by filters. Three to four filters were selected and analyzed for each month trying to select the same sampling periods for all stations. The atmospheric $PM_{10}$

concentrations during the sampling periods in the five stations range from 2 to 138 µg.m$^{-3}$, covering typical values of both background and particulate pollution periods.  Blank filters were analyzed by RSQA and showed no significant amount of S (below the detection limit of 0.2 mg.L$^{-1}$) and generated no sulfate when we followed the chemistry method that we are describing below.

### 2.2    Sulfur multi-isotope analysis

Before analyzing the S multi-isotope compositions, speciation of S in the aerosol samples needed to be addressed to understand the form(s) under which S was present (i.e. sulfates, sulfur and/or sulfide; Longo et al. (2016)). Neither elemental sulfur nor



sulfide was detected in our selected representative samples (n=5) indicating that S probably only occurs as sulfates. For each filter sample, a ~ 5 cm x 5 cm piece was cut and inserted into a reaction vessel heated at 180°C with 20 mL of Thode solution, a mixture of hydrochloric, hydroiodic and hypophosphorous acids (Thode et al., 1961), for 1.3 hours to quantitatively reduce sulfate into $H_2S$. The formed gases were purged from the vessel using nitrogen gas, bubbled through deionized water and

subsequently passed through a 0.3 M silver nitrate ($AgNO_3$) solution to form silver sulfide ($Ag_2S$). This solid $Ag_2S$ was then rinsed twice with Millipore water and dried at 70°C overnight. $Ag_2S$ was then loaded into an aluminum foil, weighted and degassed under vacuum.

$Ag_2S$ was subsequently converted to $SF_6$ by reacting with approximately 200 Torr of excess fluorine in a nickel bomb at 250°C. The produced $SF_6$ was purified using both cryogenic techniques and gas chromatography, quantified and subsequently

analyzed by dual inlet isotope ratio mass spectrometry (Thermo-Fisher MAT-253) where m/z = 127, 128, 129 and 131 ion beams were monitored. For S amounts down to 0.1 µmol, samples were analyzed using a micro-volume device (Au Yang et al., 2016). For S amounts smaller than 0.1 µmol, only the $\Delta^{33}S$-values are reported, as the intensity of the $^{36}S$ is then too small to be properly analyzed.

The $\delta^{34}S$-values were measured against our in-house $SF_6$ tank that had been previously calibrated with respect to the IAEA-

S1, IAEA-S2 and IAEA-S3 international standards and expressed versus V-CDT assuming a $\delta^{34}S_{S1}= -0.3‰$ vs V-CDT isotope composition. To express our $\Delta^{33}S$ and $\Delta^{36}S$ data with respect to V-CDT, we anchored our data using CDT-data measured previously in our laboratory following Defouilloy et al. (2016). No further corrections were carried out, other than normalization of the data to CDT. $\Delta^{33}S$ and $\Delta^{36}S$ IAEA-standards were within values reported elsewhere (Labidi et al., 2012;Defouilloy et al., 2016;Au Yang et al., 2016). Our analysis (n = 8) of IAEA-S1 standard yielded: $\delta^{34}S = -0.29 \pm 0.04‰$,

$\Delta^{33}S = 0.080 \pm 0.010‰$ and $\Delta^{36}S = -0.852 \pm 0.085‰$ vs CDT. Analysis of IAEA-S2 standard (n = 8) gave: $\delta^{34}S = 22.49 \pm 0.26‰$, $\Delta^{33}S = 0.025 \pm 0.005‰$ and $\Delta^{36}S = -0.196 \pm 0.223‰$ vs CDT. Analysis of IAEA-S3 standard (n = 8) gave: $\delta^{34}S = -32.44 \pm 0.30‰$, $\Delta^{33}S = 0.069 \pm 0.023‰$ and $\Delta^{36}S = -0.970 \pm 0.277‰$ vs CDT. Analyses of the international sulfate standard NBS-127 was also performed and gave a $\delta^{34}S$ of $20.8 \pm 0.4‰$ ($2\sigma$; n = 12), consistent with the $20.3 \pm 0.4‰$ value reported by the IAEA.

## 2.3    Chemical analysis

Concentrations of selected soluble inorganic species ($Na^{2+}$, $NH_4^+$, $K^+$, $Ca^{2+}$, $Mg^{2+}$, $NO_3^-$, $SO_4^{2-}$, $Cl^-$, $F^-$, $PO_4^{3-}$) were measured by ion chromatography (Professional IC 850 by Metrohm®) after extraction of another 3 cm x 3 cm filter piece in 30 mL Milli-Q water (Paris et al., 2010). Detection limits for ionic species are usually in the order of 5 µg.L$^{-1}$, i.e. 0.1 ng.m$^{-3}$

considering our sampling and extraction protocol.





## 3    Results

### 3.1    Multiple sulfur isotopic compositions

#### 3.1.1    Description

Sulfur multi-isotope compositions in aerosol sulfates from stations 03, 06, 13, 50 and 98 are reported in Table S1, S2, S3, S4,
and S5 respectively. For station 03, $\delta^{34}$S-values vary from 2 to 8 ‰, $\Delta^{33}$S-values from -0.006 to 0.208 ‰ and $\Delta^{36}$S-values from
0.9 to -0.7 ‰. For station 06, $\delta^{34}$S-values vary from 2 to 11 ‰, $\Delta^{33}$S-values from -0.075 to 0.319 ‰ and $\Delta^{36}$S-values from -
0.8 to 0.7 ‰. For station 13, $\delta^{34}$S-values vary from 2 to 11 ‰, $\Delta^{33}$S-values from -0.080 to 0.209 ‰ and $\Delta^{36}$S-values from -0.5
to 0.8 ‰. For station 50, $\delta^{34}$S-values vary from 3 to 11 ‰, $\Delta^{33}$S-values from -0.018 to 0.316 ‰ and $\Delta^{36}$S-values from -0.5 to
0.8 ‰. For station 98, $\delta^{34}$S-values vary from 2 to 8 ‰, $\Delta^{33}$S-values from -0.022 to 0.341 ‰ and $\Delta^{36}$S-values from -1 to 1.7‰.
The station 98 (i.e. westernmost station likely less influenced by local anthropogenic emissions) presents $\delta^{34}$S-values ranging
from 2 to 8‰ while other stations, likely impacted by local anthropogenic sources (stations 03, 06, 13 and 50), have $\delta^{34}$S
ranging from 2 to 12‰. Stations 6, 13 and 50, i.e. more influenced by highways, the downtown and maritime traffic
respectively, typically present the highest $\delta^{34}$S-values and display a similar general trend with the seasonality. All the station
(03, 06, 13, 50 and 98) present similar range of $\Delta^{33}$S-values ranging from -0.075 to 0.341‰. On the contrary, the $\Delta^{36}$S-values
for stations around Montreal downtown (from -0.8 to 0.8‰) are in the isotope range found for station 98 (from -1 to 1.7‰).

#### 3.1.2    Evidence of both seasonality

In order to reveal the seasonal variations of the S multi-isotope compositions for each station, a locally weighted scatter plot
smoothing (LOWESS) was applied (Figure 1). All stations present seasonal $\delta^{34}$S variations that follow a similar pattern (Figure
1): low $\delta^{34}$S-values from early winter on, followed by significant $^{34}$S enrichment during spring. Then, the isotope compositions
decrease from the summer to autumn, back to the early winter values. However, isotope differences in the $\delta$ $^{34}$S range are
observed between sites.



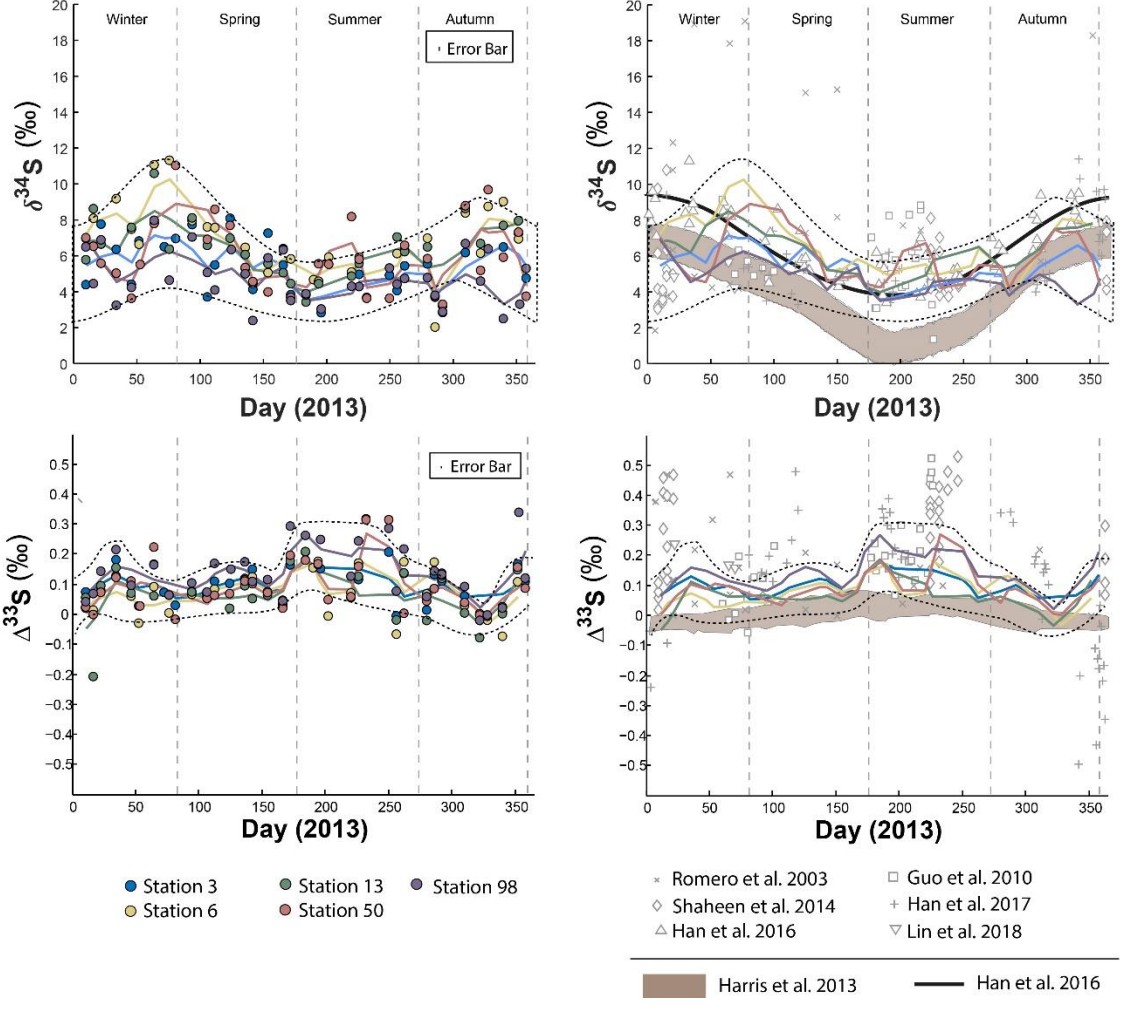

**Figure 1 : Sulfate $\delta^{34}S$ and $\Delta^{33}S$ variations over time in Montreal PM$_{10}$ aerosols. Grey bands represent the modelled sulfate isotope characteristics if SO$_4$ is formed following the oxidation pathways proposed by (Harris et al., 2013a).**

5    During spring, all stations display a similar $\delta^{34}S$ pattern characterized by an enrichment in $^{34}S$, with station 06 yielding the highest $\delta^{34}S$ (up to 10‰), followed by stations 03, 50 and 13 with isotope compositions up to 8‰. The station 98, in contrast, presents the lowest $\delta^{34}S$-values during that period, down to 5‰. However, the difference in $\delta^{34}S$-values between the stations 03, 06, 13 and 50 becomes less significant the rest of the year, yielding a near constant $\delta^{34}S$-value of 8‰ during both summer and winter. During winter, the station 98 still shows lower $\delta^{34}S$-values (around 5‰).

10    The seasonal variations proposed by Harris et al. (2013a) for rural aerosols from continental northern mid-latitudes (45°N) is also represented (brown field) in Figure 1. This model (i.e. second model described in the introduction section) relies upon the isotope seasonal variations induced by three major oxidation pathways (OH, H$_2$O$_2$, O$_2$+TMI) involved in the oxidation of 43%



of the atmospheric $SO_2$, the other 57% being dry-deposited. As Montreal latitude (45°N 73°W) is similar to the one considered by Harris et al. (2013a) both datasets may be compared. The seasonality trend documented in Beijing by Han et al. (2016) (thick black line in Figure 1) highlights high $\delta^{34}S$-values up to 10‰ during spring and winter while summer is characterized by low $\delta^{34}S$-values down to 4‰. According to Han et al. (2016), following the first model in the introduction section (i.e. S

isotope systematic as a direct tracer of emission sources), this seasonality would reflect changes in the respective contributions of sources of atmospheric sulfate during different times of the year rather than changes in the $SO_2$ oxidation chemical pathways. While our data are consistent at the first order with the seasonality highlighted for urban aerosols (Han et al., 2016) and the seasonality modelled for rural aerosols (Harris et al., 2013a) for the period bridging from the end of spring to the end of autumn, they show a significant deviation with a $\delta^{34}S$-decrease between early winter and early spring in Montreal which is neither

predicted by Harris et al. (2013a) model nor observed in Beijing.

The $\Delta^{33}S$ measured at the five stations also show seasonal variations displaying $^{33}S$ enrichment up to 0.3‰ during early winter and summer and $\Delta^{33}S \sim 0$‰ in spring and autumn (Figure 1 ; dotted lines). Despite that stations 03, 06, 13 and 50 present a similar $\Delta^{33}S$-range than the station 98, mean $\Delta^{33}S$-values from the LOWESS show that station 98 is characterized by the highest $\Delta^{33}S$-value (0.143‰) compared to the others (ranging from 0.064 to 0.101‰). Lowest $\Delta^{33}S$-values are recorded along

the year by stations 03, 06 and 13. In contrast, most of the stations present similar $\Delta^{36}S$-values, except for station 98 that yields the largest range of $\Delta^{36}S$ from -1 to 1.7‰ (Table S5) so no seasonal $\Delta^{36}S$-variations are highlighted (Fig. S1). The model proposed by Harris et al. (2013a) also suggested a seasonal variation (brown area in Figure 1) for rural aerosols with a maximum $\Delta^{33}S$-value of 0.05‰ during summer and a minimum of -0.05‰ in winter. Clearly, this model cannot explain the larger range of S isotope compositions observed in Montreal.

**3.1.3    Comparison with the literature**

S-isotopes data have been previously obtained on aerosols collected in the United States of America (Romero and Thiemens, 2003;Shaheen et al., 2014) : La Jolla (32°N and 117°W, rural environment), Bakersfield (35°N and 119°W, urban environment), White Mountain (37°N, 118°W, rural environment) and in China (Guo et al., 2010;Han et al., 2017;Lin et al., 2018b): Beijing (39°N and 116°E, urban environment), Guangzhou (23°N and 113°E, urban environment) and Xianghe (39°N

and 116°E, urban environment).

The $\Delta^{33}S$-values in Montreal aerosols share common characteristics with most of the available data (Guo et al., 2010;Romero and Thiemens, 2003;Shaheen et al., 2014;Lin et al., 2018b) with high $\Delta^{33}S_T$-values (i.e. the $\Delta^{33}S$ measured and non-corrected for sea-salt) occurring during both summer and winter sampling periods (open gray marks in Figure 1). Collectively these data differ from the study by Han et al. (2017) who recently reported the first negative $\Delta^{33}S$-values, down to -0.6‰, in Beijing.



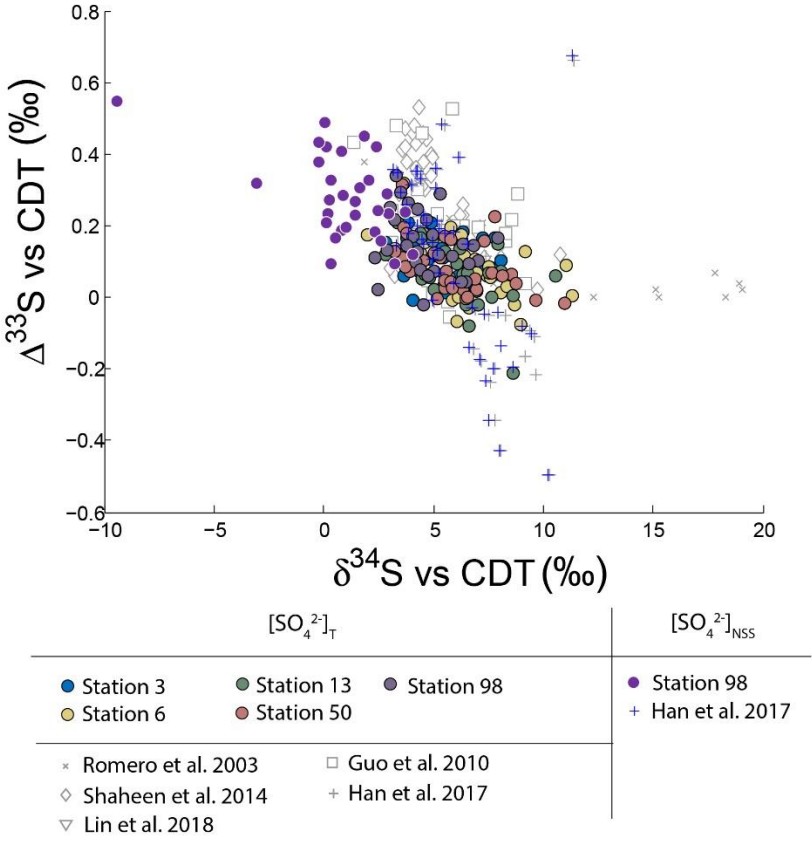

**Figure 2 : Variations of $\Delta^{33}$S as a function of $\delta^{34}$S in sulfates from PM$_{10}$ aerosols collected in Montreal. Light grey dots are compiled from (Guo et al., 2010;Han et al., 2017;Romero and Thiemens, 2003;Shaheen et al., 2014). Dark grey dots represent the NSS-s-corrected multiple sulfur isotope compositions.**

Neglecting seasonality, it can be noted that worldwide urban aerosols present a range of $\delta^{34}$S-values varying mostly from 0 to 20‰ with a mean value of $6 \pm 5$‰ ($2\sigma$) and $\Delta^{33}$S-values varying from -0.6 to 0.7‰ with a mean value of $0.17 \pm 0.4$‰ ($2\sigma$) (Figure 2). The model presented by Harris et al. (2013a) cannot explain these $\delta^{34}$S-values as it predicts isotope variations ranging between 1 and 7‰. Also, Montreal samples present a wider range of $\Delta^{36}$S-values (from -1 to 2‰) compared to the one (-2 to 0‰) reported in the literature with significant positive isotope compositions for half of our samples (Figure 3). The observation of positive $\Delta^{36}$S-values are surprising but do not come from analytical issue as these are observed in several samples, some of them having been duplicated. This represents, to our knowledge, the first positive $\Delta^{36}$S-values reported for urban aerosols. To date, Baroni et al. (2008) is the only other study reporting both positive $\Delta^{33}$S and slightly positive $\Delta^{36}$S-values for one volcanic sample that the authors considered to result from mass-dependent fractionation. Recently, the $\Delta^{36}$S-values have been suggested to be decoupled from the $\Delta^{33}$S-values, the $\Delta^{36}$S-values in aerosols being explained by combustion while the $\Delta^{33}$S-values would reflect another atmospheric process e.g. input of stratospheric sulfates (Lin et al., 2018b). Both combustion process and stratospheric inputs could explain the background tropospheric sulfates in China, which is



characterized by positive $\Delta^{33}S$ and negative $\Delta^{36}S$-values (Lin et al., 2018b), but they cannot account for the both positive $\Delta^{33}S$ and $\Delta^{36}S$-values measured in Montréal. Up to now, only the experimental photooxidation has been shown to produce both positive $\Delta^{33}S$ and $\Delta^{36}S$-values (slope ~ 0.64) (Whitehill and Ono, 2012), which could highlight the implication of this process in the samples in Montréal.

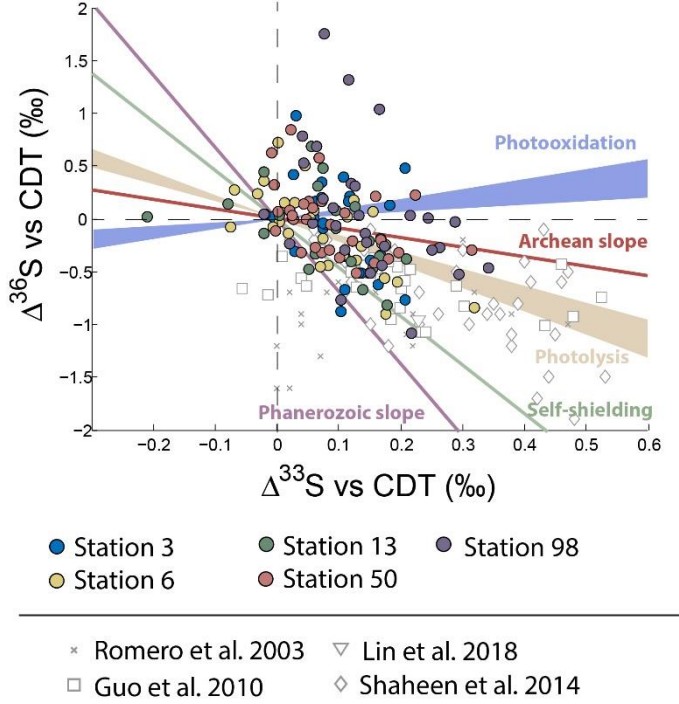

**Figure 3 : Variations of $\Delta^{33}S$ as a function of $\delta^{34}S$ in sulfates from PM$_{10}$ aerosols collected in Montreal. Archean and Phanerozoic slopes taken from the literature are also reported.**

### 3.1.4    Potential sea –salt contribution

The concentrations of Na$^+$ in aerosols from station 98 ranged from 0.08 to 11.85 µg.m$^{-3}$ and those of SO$_4^{2-}$ from 0.04 to 2.42 µg.m$^{-3}$ (Table S5). The presence of detectable Na$^+$ concentrations in all samples from this station may either reflect the local use of NaCl deicing road salts during winter (Zinger and Delisle, 1988) and/or the contribution of sea-salt aerosols. If deicing NaCl was the source of aerosol Na, we would expect higher Na$^+$ amounts in aerosols during winter. Yet, Na$^+$ concentrations do not show any significant variations along the year, precluding the input from deicing salts highlighting the contribution of sea-salt in the observed Na$^+$ concentrations.

It is worth mentioning that among the aerosols collected at station 98, two of them show a Na/Cl ratio (~1) similar to sea-salt sulfates (~0.6). The other samples show very high Na/Cl ratios (>10). This shows a depletion of Cl in most of our samples, which could be explained by a reaction between NaCl and sulfate/nitrate that forms NaSO$_4$ or NaNO$_3$, a process also known as the "aging of sea-salt particles" (Sinha et al., 2008;Song and Carmichael, 1999). The aging process involves the exposure

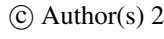



of secondary aerosols precursors, which would form sulfates in a marine environment. In this context sulfates could either result from the oxidation of $SO_2$ formed by the oxidation of DMS or from the oxidation of anthropogenic $SO_2$ (Harris et al., 2012c). DMS is characterized by a mean $\delta^{34}S$-value of 18‰ and a mean $\Delta^{33}S$-value of 0.03‰ (Oduro et al., 2012) and anthropogenic $SO_2$ by variable $\delta^{34}S$-values and a mean $\Delta^{33}S$-value of -0.018‰ (Lin et al., 2018b).

Thus, any contribution from aged-sea salt to the sulfates budget would decrease the $\Delta^{33}S$-values measured in aerosols. It is thus essential to correct for these contributions. Considering that the different major oxidants in the marine environment ($H_2O_2/O_3$, $O_2$+TMI, OH) would form sulfates with similar $\Delta^{33}S$-anomalies than the initial $SO_2$ and that the $\Delta^{33}S$-values of sea-salt sulfates, DMS and anthropogenic $SO_2$ are very similar (~ 0.05‰), we corrected our $\Delta^{33}S$-values using the 0.05% $\Delta^{33}S_{SS}$-values reported by previous studies (Labidi et al. (2012);Ono et al. (2006a);Ono et al. (2006b)). Both DMS and sea-salt sulfate

are characterized by similar $\delta^{34}S$-values. The $\delta^{34}S$ of anthropogenic $SO_2$ is more variable but the corresponding correction for the $\delta^{34}S$-value is of little importance and will thus not change the conclusion we are making following the study of the $\Delta^{33}S_{NSS}$-values. Ultimately, we thus applied an isotope balance equation to calculate the $\delta^{34}S_{NSS}$ and $\Delta^{33}S_{NSS}$ isotope compositions corresponding to the non-sea salt (NSS) sulfates fraction for each aerosol sample using average $\delta^{34}S_{SS}$ (= 22‰) and $\Delta^{33}S_{SS}$ values (0.05‰) for sea-salt sulfate. A detailed description and results ($\delta^{34}S$, $\Delta^{33}S$ and percentages) are reported in Table S6. A

consequence of this correction is that it both decreases the $\delta^{34}S$-values of the aerosol samples and increases their corresponding $\Delta^{33}S$-values. Results show that in average sulfates sampled at station 98 include ~ 80 ± 10% of NSS-sulfates that are characterized by $\Delta^{33}S_{NSS}$ values varying from 0.010 to 0.550 ± 0.1‰ with a mean value of 0.280 ± 0.118‰. Two $\Delta^{33}S_{NSS}$ values from station 98, calculated at 2.197 and -0.723‰ (01/22 and 12/06, respectively; Table S6), were not considered as one is characterized by a low ~30% NSS-sulfate concentration making it very sensitive to the SS-correction and the other one

present a too high concentration of Na which induces a negative percentage (-153%) of NSS (Table S6). As the SS-contribution appears constant along the year, this correction does not affect the seasonality pattern highlighted in Sect. 3.1.2.. Other stations (P03, 06, 13, 50) show similar NSS contributions (~ 80%; Table S7) and $\Delta^{33}S_{NSS}$-values that also range between 0 and 0.5 ± 0.1‰ but station 98 still presents the highest mean $\Delta^{33}S_{NSS}$-values.

It is worth noting that the highest $\Delta^{33}S_T$-value measured in Montreal (0.35‰) is lower than the one reported in Xianghe and

Beijing (China; 0.5‰; Guo et al. (2010);Han et al. (2017)). This difference may be attributed to significantly different sea-salt contributions: 20% in Montreal compared to 1% in Beijing (Han et al., 2017). Due to this low 1% sea-salt contribution in Beijing, the corresponding corrected isotope compositions NSS-s (blue marker in Figure 2) are not significantly affected. For Montreal, the corrected $\Delta^{33}S_{NSS}$ reach values as high as ~0.5 ± 0.1 ‰, therefore similar in magnitude to other studies. The fact that sulfate aerosols from Beijing and Montreal display the same highest $\Delta^{33}S$ may highlight a common reaction scheme. Thus,

our discussion will be based on the assumption of a common process explaining the high $\Delta^{33}S$-values observed in the two cities.



## 4    Discussion

### 4.1    Anthropogenic emission, $\Delta^{33}$S-values and seasonality

Aerosols collected at stations likely impacted by local emission sources (i.e. stations 03, 06, 13 and 50) present the lowest $\Delta^{33}$S-values (~ -0.01‰) and the highest $\delta^{34}$S-values (up to ~ 12‰) compared to station 98 (less influenced by anthropogenic

emissions). This suggests that local emissions in Montreal are characterized by $\delta^{34}$S-values up to 12‰ and mean $\Delta^{33}$S-values close to 0‰, which implies that the high $\Delta^{33}$S-anomalies with lower $\delta^{34}$S are transported to rather than produced in Montreal. Local anthropogenic sources could then isotopically impact these imported aerosol sulfates by decreasing their $\Delta^{33}$S-values towards 0‰. This is consistent with Lee et al. (2002) who showed the ability of these primary aerosols, resulting from the combustion process, to decrease $\Delta^{33}$S-values towards 0‰, as they are characterized by zero to slightly negative $\Delta^{33}$S-values

down to -0.2‰. Secondary sulfates formed by $SO_2$ oxidation within cities by the main oxidation pathways (OH, $O_2$ + TMI, $NO_2$, $H_2O_2$, $O_3$)  would not generate significant MIF and would also lead to decrease the $\Delta^{33}$S-value of imported aerosols. (Harris et al., 2013a;Au Yang et al., 2018).

This contrasts with the interpretation where the negative $\Delta^{33}$S-values (down to -0.6‰) measured during winter in Beijing would relate to anthropogenic sources, in particular those generating incomplete combustion processes (Han et al., 2017). The

authors mostly rely on data showing that primary aerosols are characterized by negative $\Delta^{33}$S-values but only down to -0.2‰ (Lee et al., 2002). However, Han et al. (2017) interpretation would predict: i) a seasonality with negative $\Delta^{33}$S-values down to -0.6‰ during winter as a result from increased coal and wood burning and ii) a gradient in the $\Delta^{33}$S-values from the outer towards the inner city with isotope shifting from ~ 0‰ to negative $\Delta^{33}$S-values. The first point contrasts with our data in Montreal and as for the second point we are actually observing a trend but with a shift from positive to zero-values. It comes

that based on the available data, S emissions from anthropogenic activities can neither explain the $\Delta^{33}$S seasonality nor the highest $\Delta^{33}$S-values up to 0.5‰ measured in urban aerosols.

Our conclusion implies that sulfur isotopes budget in rural environments is not only governed by $SO_2$ oxidation by atmospheric $O_2$+TMI, $H_2O_2$, $O_3$ $NO_2$ and OH: this point is discussed further using results from station 98 given it is the one being less affected by emissions from local anthropogenic activities.

### 4.2    Input of stratospheric sulfates, $\Delta^{33}$S-values and seasonality

To date, sulfates samples trapped in the Antarctica ice are showing the most extreme non-zero $\Delta^{33}$S-values with negative and positive $\Delta^{33}$S-values down to -2‰ and up to 1‰, respectively (Baroni et al., 2008;Baroni et al., 2007;Bindeman et al., 2007;Savarino et al., 2003;Shaheen et al., 2014;Gautier et al., 2018). Their formation results from the photochemical oxidation of atmospheric $SO_2$. This is because $SO_2$ possesses two dominant absorption bands in the ultraviolet region: one at 190–220

nm (photolysis) and the other one at 250–330 nm (photooxidation), which are able to create high $\Delta^{33}$S-values up to 15‰ (Farquhar et al., 2000;Farquhar et al., 2001;Whitehill et al., 2015;Whitehill and Ono, 2012;Whitehill et al., 2013). While produced under distinct $O_2$ levels, this process has been suggested to account for sulfur multiple isotope signatures of both



Archaean sediments (Whitehill and Ono, 2012) and modern aerosols in Antarctica (Savarino et al., 2003;Hattori et al., 2013;Baroni et al., 2007;Gautier et al., 2018).

Given the similar $\Delta^{33}$S-values between urban ($\leq$ 0.5‰) and Antarctica aerosols ($\leq$ 2‰), the $\Delta^{33}$S-values of urban aerosols could result from inputs of stratospheric sulfate aerosols, supposedly carrying significant $\Delta^{33}$S anomalies into the troposphere

(Guo et al., 2010;Romero and Thiemens, 2003;Lin et al., 2018a;Lin et al., 2018b). However, according to the HYSPLIT three-days back-trajectory analysis carried out for each of our samples in Montreal (Figure 4), the probability of injecting stratospheric air masses into the troposphere is relatively low, the HYSPLIT model predicting that only ~ 10% of the aerosols were coming from altitudes higher than 500 m. Furthermore the stratosphere troposphere exchange (STE) in the northern hemisphere is preferentially located in Northern Pacific and Northern Atlantic (Boothe and Homeyer, 2017;Gettelman et al.,

2011;Sprenger and Wernli, 2003) and occurs during a period less than 80 days per year (Boothe and Homeyer, 2017). This again is not favoring the hypothesis that injection of stratospheric sulfate into Montreal urban atmosphere can explain the occurrence of non-zero $\Delta^{33}$S-values. This is consistent with the study of Lin et al. (2016) who also estimated a very low (~ 1 %) input of stratospheric $SO_4$ in their study. Considering 1% associated with a maximal $\Delta^{33}$S anomaly of 10 ‰ for stratospheric sulfates (Ono et al., 2013) this would result in a $\Delta^{33}$S-value of only 0.1‰ in tropospheric sulfates. Finally, based on the

Antarctic sulfate isotope record, the $\Delta^{33}$S anomalies of the stratospheric sulfates produced by photochemical processes would only occur when high $SO_2$ concentrations are reached, typically following stratospheric volcanic eruptions (Ono et al., 2013;Baroni et al., 2007;Savarino et al., 2003;Baroni et al., 2008;Martin, 2018). Ultimately, inputs of stratospheric aerosol sulfates can thus hardly explain the sustainable urban aerosol $\Delta^{33}$S anomalies that are observed worldwide.

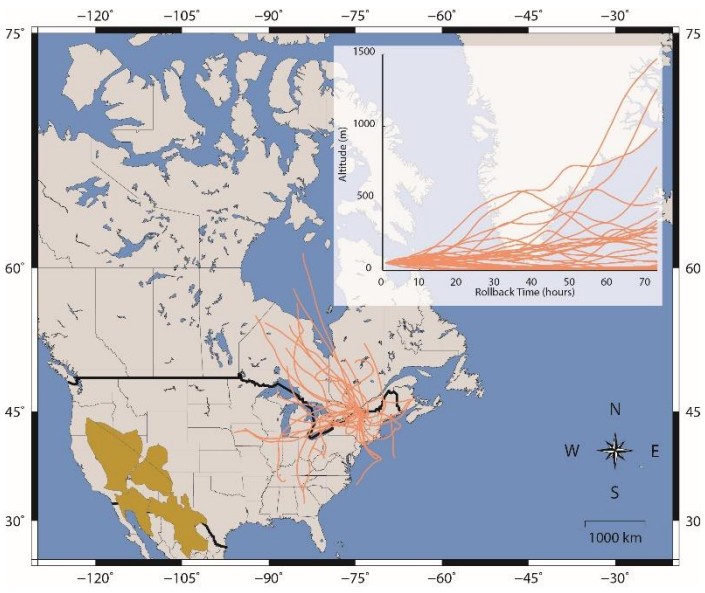

**Figure 4 : Three-days back-trajectories modelled using the HYSPLIT software for the sampling stations in Montreal. Back trajectories are calculated using an initial height of 50 m above the sea level. American deserts are also shown in brown and include the Great Basin, the Mohave desert, the Chihuahua desert and the Sonoran desert.**



### 4.3    Formation of secondary sulfates, $\Delta^{33}$S-values and seasonality

Romero and Thiemens (2003) highlighted a negative dependence between $\Delta^{33}$S-values and the aerosol size, down to typical secondary aerosol sub-micron sizes (Seinfeld and Pandis, 2012) associated with a $\Delta^{33}$S increase with values up to 0.5‰, similar to values measured among aerosols. The fact that the main oxidants OH, $H_2O_2$, $O_2$+TMI, $O_3$ and $NO_2$ cannot produce $\Delta^{33}$S-values higher than ± 0.2‰ (Au Yang et al., 2018;Harris et al., 2013a) arises the need for another oxidation. It comes that the only process that could preserve the non-zero $\Delta^{33}$S signatures in urban aerosols after mixing with aerosols that underwent the main near mass-dependent oxidation reactions highlighted above would be an oxidation associated with high $\Delta^{33}$S-values. We explore in the following a non-exhaustive series of reactions which we believe require special attention and discussion.

### 4.3.1    Oxidation by the Stabilized Criegee Intermediate (SCI)

Previous studies suggested that $SO_2$ oxidation by the Stabilized Criegee Intermediate (SCI) may represent a reliable possibility (Mauldin III et al., 2012;Sarwar et al., 2014;Boy et al., 2013;Ye et al., 2018;Sipilä et al., 2014). The SCI oxidation pathway is important in rural environments (characterized by higher VOC/NOx ratios compared to the urban atmosphere) and would mitigate the discrepancy between modelled and measured $SO_4$ concentrations (Sarwar et al., 2013;Sarwar et al., 2014). Boy et al. (2013) showed, using a one-dimensional model that SCI reaction with $SO_2$ may represent 40% of the total sulfuric acid production during winter in the atmospheric boundary layer above the forest canopy, while this contribution varies between 15 and 20% for the other seasons. Thus, although the contribution of the SCI oxidation pathway might be low (representing 0.69% of the $NO_2$ oxidation pathway; (Cheng et al., 2016)), it might represent a viable alternative to be considered in the future, notably in winter. If this SCI oxidation pathway generates high $\Delta^{33}$S-anomalies, it would account for the observation that $\Delta^{33}$S-values in winter are more important at station 98 (i.e. background) than at stations located downtown. In this case, the high anomalies found during summer for the station 98 could be related to another process.

### 4.3.2    SO₂ photooxidation

Another possibility, based on the study by Whitehill and Ono (2012) and Ono et al. (2013) could be the still unexplored $SO_2$ photooxidation in the troposphere. The wavelength range of the actual actinic flux at sea level (in the troposphere) varies from 300 to 2500 nm (Eltbaakh et al., 2011) because ozone ($O_3$) absorbs the bulk of the solar UV radiation in the 290-320 nm region (Molina and Molina, 1986). The overlapping region shows that tropospheric $SO_2$ absorption can only occur within a narrow wavelength range (typically 320 to 330 nm ; Calvert et al. (1978)) that still leave room for S-MIF to be produced. Although the cross-section in this overlapping region is small (~ 0.2 x 10$^{-18}$ cm² ; Blackie et al. (2011)), experimental studies need to be performed to estimate the S multiple isotope compositions of sulfates formed under these particular conditions. To our knowledge, there is no data available yet within this narrow wavelength range (Farquhar et al., 2001;Whitehill et al., 2013;Whitehill et al., 2015;Whitehill and Ono, 2012;Danielache et al., 2012). This process would likely be more significant during summer but would hardly explain the high $\Delta^{33}$S-values observed in winter.



### 4.3.3 Aging of sea-salt sulfate

Chemical analyses of our samples show a correlation between sulfate and Na concentrations, suggesting sodium sulfate is probably the main form of sulfate in our samples. The formation of sodium sulfate result from the $SO_2$ oxidation coming either from direct emission or DMS oxidation on sea-salt aerosols surface (Harris et al., 2012d;Sinha et al., 2008). Thus, the aging process, which induces a Cl depletion, might explain such $\Delta^{33}S$-values. In this perspective and according to the hypotheses in Sect. 3.1.4, as for urban aerosols sulfates in Montréal, a Cl depletion should also be observed in the aerosols in Beijing. However, the contribution of the sea-salt sulfates in those samples represents only 1% of the total aerosols without showing a Cl depletion which rules out this hypothesis and thus indicate the occurrence of another oxidation process.

### 4.3.4 Photooxidation of SO₂ in presence of mineral dust

Finally, we suggest that the photooxidation of $SO_2$ in the presence of mineral dust may represent an alternative way to generate these non-zero $\Delta^{33}S$-values in sulfate aerosols. Several studies showed that mineral dusts promote sulfate formation by either heterogeneous $SO_2$ oxidation (i.e. oxidation of the $SO_2$ adsorbed onto the mineral dust by several possible oxidants including $O_3$, $H_2O_2$, $NO_2$) or by gaseous oxidation by OH· or peroxide radical anion $O_2\cdot$ formed by UV radiation on semi-conducting metal oxides such as $Al_2O_3$, $Fe_2O_3$ and $TiO_2$. (Yu et al., 2017;He et al., 2014;Dupart et al., 2012;George et al., 2015;Usher et al., 2003;Zhao et al., 2018;Ma et al., 2018). Thus, depending on the oxidation pathway, mineral dust could promote either the increase of the aerosol size or the formation of new particles, respectively. This oxidation pathway would also mitigate the discrepancy between modelled and measured $SO_4$ concentrations (Fu et al., 2016).

Our chemical analyses do not indicate any contributions of dust particles in Montreal as: i) samples present low Fe/Al ratios (typically 0.05), distinct from the ones characterizing desert dusts, which vary from 0.48 to 1.74 (Formenti et al., 2011) and ii) modelled back trajectories (in Fig. 4) indicate that the air masses reaching the city are unlikely influenced by the western deserts. Still, we suggest that oxidation of $SO_2$ in presence of mineral dust could still occur.

Asian deserts (Cottle et al., 2013;McKendry et al., 2001) and to a lesser extent the Sahara desert (Chin et al., 2007) have been shown to represent the main sources of mineral dust affecting the North American continent, with events mainly recorded during spring and summer (Zhao et al., 2006;Prospero, 1996). The seasonality of long-ranged transport of mineral dust over the North Atlantic is concomitant with the period when the lowest $\delta^{34}S$-values and highest $\Delta^{33}S$-values are measured. This highlights a potential link between the S isotope variations and mechanisms involving mineral dust. It is particularly interesting to note that the transport of $SO_2$ from the east Asian major sources to North America is typically observed (Clarisse et al., 2011) with a mixing of sulfates on mineral dust reported over the Northern Hemisphere continents (Bauer and Koch, 2005). This suggests that the transported mineral dust is typically coupled with sulfates or in mixing with $SO_2$. However, models shows a decline in the dust/total sulfate ratio during trans-pacific transport due to an enhanced settling of super-micron dust particles compared to the fine ammonium sulfate (Fairlie et al., 2010), precluding the observation of mixed dust over North



America. Thus, the current knowledge of the transport and reactivity of mineral dust and $SO_2$ over North America is consistent with the photooxidation of $SO_2$ in the presence of mineral dust.

It is also worth noting that the seasonality reported in dust particles fluxes by Félix-Bermúdez et al. (2017) in Southern California displays a strong similarity with the $\delta^{34}S$ seasonality observed in our Montreal aerosols (Figure 5) with high $\delta^{34}S$-

values associated with high dust particles fluxes. Importantly, Figure 5 also highlights an anti-correlation trend between dust particle fluxes and $\Delta^{33}S$-values with low dust particle fluxes (~ 14 mg.m$^{-2}$.d$^{-1}$) associated with the high $\Delta^{33}S$-values (0.3‰). Although we are comparing two locations separated by several thousand kilometers (i.e. California and Quebec), Félix-Bermúdez et al. suggest that the seasonality they are observing is extendable to a global scale as the dust deposition rate is mainly driven by the contrasting atmospheric air mass transport taking place during the warm and cool seasons (Félix-

Bermúdez et al., 2017). This last point may again highlight a potential link between the S isotope variations and mechanisms involving mineral dust.

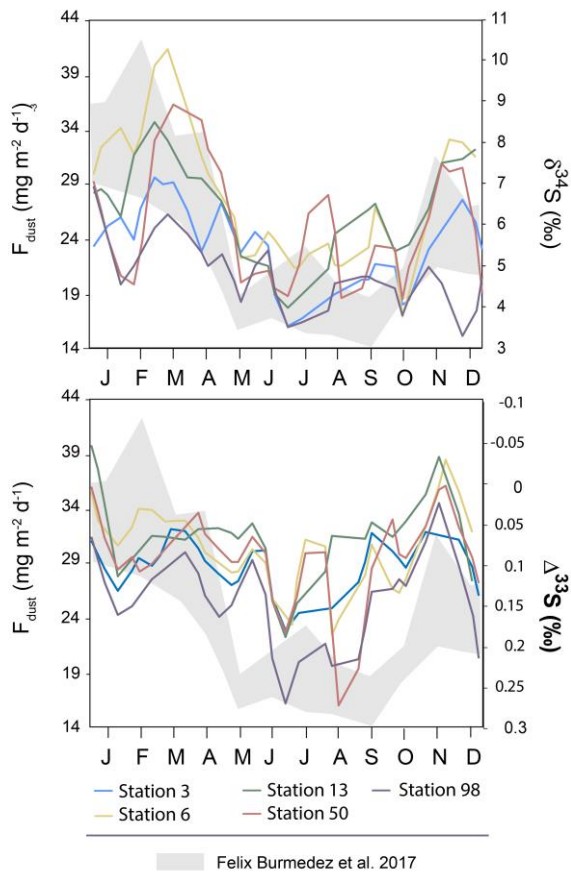

**Figure 5 : Variations of the mean $\delta^{34}S$ and $\Delta^{33}S$-values over time in sulfate aerosols collected in Montreal. The grey band represents**
**the seasonality of the dust deposition rate determined by Félix-Bermúdez et al. (2017)**




In order to confirm that the $\Delta^{33}S$ seasonality we are observing in Montreal aerosols does not reflect the signatures of primary mineral dust particles transported to the city, we analyzed desert dust samples from China, Morocco, Tunisia and Jordania. Results show $\delta^{34}S$-values varying from 5 to 13‰ but importantly no significant $\Delta^{33}S$-anomaly (Table S8).

We suggest that the $SO_2$ photooxidation reaction may occur at the dust surface and, by oxidizing the surrounding $SO_2$ into

5 sulfates, it would deplete the resulting $SO_4$ in $^{33}S$ and by mass balance, leave the residual $SO_2$ enriched in $^{33}S$ (Figure 6). Sulfates associated to the dust would be characterized by negative $\Delta^{33}S$-values and will be deposited while the residual atmospheric $SO_2$ (i.e. characterized by positive $\Delta^{33}S$-values) would be transported to Montreal. The transported $SO_2$ enriched in $^{33}S$ would then be oxidized into sulfates in Montreal vicinity through the major oxidation pathways ($O_2$+TMI, $H_2O_2$, $O_3$, OH, $NO_2$).

This oxidation pathway may occur at larger scale and may also be involved in the formation of urban aerosols reported in the literature. Indeed, urban aerosol sulfates previously studied in La Jolla, Bakersfield and White Mountain in the United States, and in Xianghe and Beijing may also be influenced by the Asian mineral dust.

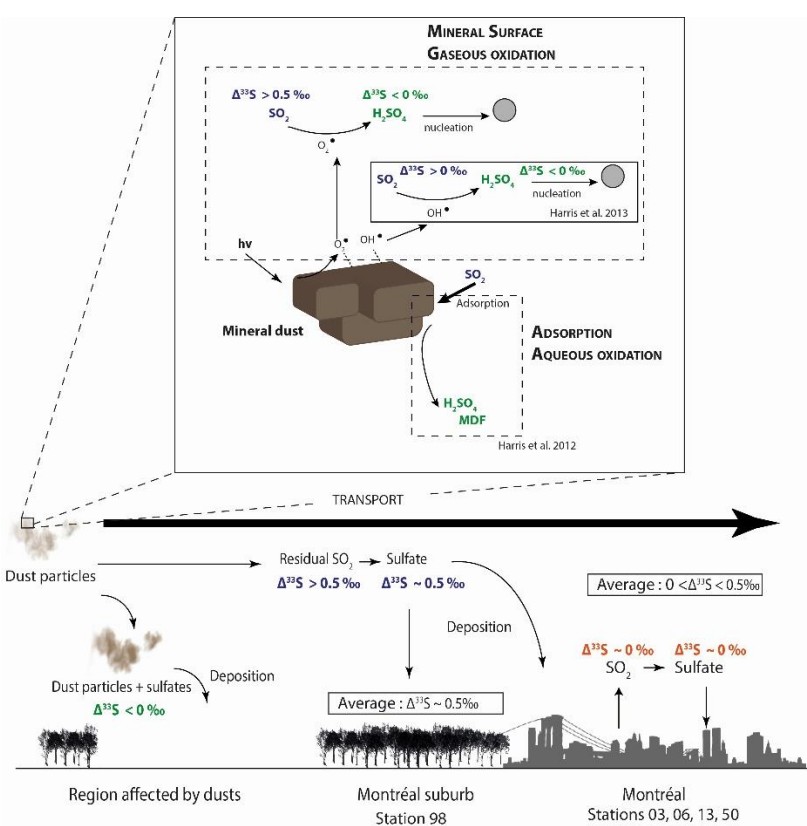

**Figure 6 : Scheme of the reaction mechanisms leading to the formation of sulfates (modified from Dupart et al. (2012)) driven by airborne mineral dust. The $^{33}S$ enriched residual SO₂ is supposedly transported to cities while the $^{33}S$ depleted sulfates is deposited along with dust particles in the rural environment. The dotted lines would depend on the meteorological conditions, which could transport dust particles and sulfates into the city.**




The exact photochemical mechanism which would be responsible for that relation remains speculative but some reactions can be highlighted and discussed. The recently proposed oxidation of $SO_2$ by $NO_2$ on mineral dust (Ma et al., 2018) is unlikely because $NO_2$ is predominant in the urban environment, i.e. at odds with the present evidence. Heterogeneous oxidation of $SO_2$ is likely to induce a mass dependent fractionation on $^{33}S$ (Harris et al., 2012a) while $SO_2$ oxidation by OH radicals produces

sulfates with negative $\Delta^{33}S$-values down to -0.15‰ (Harris et al., 2013a), leaving the remaining $SO_2$ with positive $\Delta^{33}S$-values. Among other reactions, $SO_2$ oxidation by the $O_2\cdot$ peroxide radical anion is another possibility that has not been isotopically characterized yet. Although this mechanism would be different from the one proposed by  Harris et al. (2012a), observing a $^{34}\alpha$ different from 1 suggests that the $SO_2$ oxidation by the mineral dust is incomplete and will leave a residual $SO_2$.

Our hypothesis could explain sulfates with positive $\Delta^{33}S$-values transported to Montreal but implies that negative $\Delta^{33}S$-values

also need to be found in dust particles. This hypothesis could leave a new room to explain negative $\Delta^{33}S$-values measured  in Beijing aerosols (Han et al., 2017). Intuitively, dust particles may be transported during discrete storm episodes (Marticorena and Bergametti, 1995;Kok et al., 2012) which have been reported mostly during spring in China (Zhao et al., 2006;An et al., 2018). Accordingly to this hypothesis, negative $\Delta^{33}S$-values would be found in spring which is not the case (Han et al., 2017). In fact, five dust episodes were identified in China in 2016 (An et al., 2018) with one (March 4th) happening close to the

sampling period (March 15th to April 26th; Han et al. (2017)). However, the images recorded by the NASA satellite show that the    dust    storm    in    the    Gobi    Desert    would    unlikely    reach    Beijing    that    day (https://modis.gsfc.nasa.gov/gallery/individual.php?db_date=2016-03-11), possibly explaining why such negative values have not been measured (Han et al., 2017).

It is worth mentioning that weak wind conditions can also be responsible for a high contribution of fine (0.15-15µm) mineral

dust fraction in the total aerosol content (Golitsyn et al., 1997;Golitsyn et al., 2003;Chkhetiani et al., 2012), being also observed in Beijing during the winter haze episodes (Yang et al., 2017;Yang et al., 2018) where negative $\Delta^{33}S$-values primarily occur (Han et al., 2017). Although the contribution of sulfates from terrigeneous sources has been estimated to a maximal value of $3.84 \pm 4.40$ to $5.62 \pm 6.52\%$ deduced from $Ca^{2+}$ concentration (Han et al., 2017). However, Asian dust present a high variability of $Ca^{2+}$ concentration (the Ca/Al ratio varying from < 0.1 to 35%; Formenti et al. (2011)) reflecting that $Ca^{2+}$ is mainly a tracer

for carbonate mineral (Formenti et al., 2011). Using conventional crustal references (Fe/Al and K/Al) (Guieu et al., 2002;Wagener et al., 2008;Paris et al., 2010;Formenti et al., 2011) may help to better discuss the contribution of dust particles. In that perspective, we predict that locations characterized by low or no mineral dust inputs but still emitting sulfates (e.g. such as South Africa or Brazil ; observable at https://svs.gsfc.nasa.gov/30017) would be characterized by low or zero-$\Delta^{33}S$-values. Identically, we predict that regions highly affected by dust could be characterized by negative $\Delta^{33}S$-values. Our study also

suggests that the aerosol sulfate-coating could be characterized by negative $\Delta^{33}S$-values. This further needs to be tested by isotopically characterizing the formation of sulfate-coating in aerosols through $SO_2$ photooxidation in presence of mineral dust.



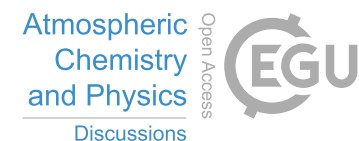

## 5    Conclusions

In this study, we determined for the first time the multiple sulfur isotope compositions of $PM_{10}$ sampled during a one-year period in Montreal at several monitoring stations disseminated within the island, each characterized by a specific environment (high-traffic highways interchange, urban background, maritime and festive activities, rural background). We demonstrated

that stations impacted by local anthropogenic emissions are characterized by higher $\delta^{34}S$ (from 2 to 12‰) and lower $\Delta^{33}S$-values that tend towards 0 ‰. The rural background station, which likely collects aerosols that are transported by upstream winds to Montreal, yielded lower $\delta^{34}S$-values (2‰) and higher $\Delta^{33}S$-values, up to 0.35‰. Our results suggest that aerosols collected within the city have their $\Delta^{33}S$-values lowered by mixing with emissions from local sources compared to aerosols sampled in the vicinity of the city. We conclude that the non-zero $\Delta^{33}S$-values we measured were rather generated upstream

the city than produced locally.

We also identified an urban seasonality for both $\delta^{34}S$ and $\Delta^{33}S$ in $PM_{10}$, with higher $\delta^{34}S$-values during early spring and autumn and higher $\Delta^{33}S$-values in summer and winter. Our results indicate that these seasonal trends cannot be explained by corresponding seasonal variations in the atmospheric concentrations of the OH, $H_2O_2$, $O_2+TMI$ and $O_3$ oxidants. In turn, we suggest that this seasonality may be better explained by either $SO_2$ oxidation by the Criegee radicals and/or $SO_2$ photooxidation

in presence of mineral dust. Still, further studies are required to isotopically characterize these latest oxidation pathways (i.e. Criegee radicals and $SO_2$ photooxidation in presence of mineral dust), which are still neglected in most current atmospheric models.

## 6    Data availability.

All data needed to draw the conclusions in the present study are shown in this paper and/or the Supplement. For additional

data related to this study, please contact the corresponding author (auyang@mail.gyig.ac.cn).

The Supplement related to this article is available online

## 7    Author contributions.

DAY conducted sulfur isotope measurements under the supervision of PC and conducted chemical composition measurements under the supervision of KD. DW provided the samples. DAY, PC, KD and DW interpreted the data. DAY wrote the paper

with contributions from all coauthors.

## 8    Competing interests.

The authors declare that they have no conflict of interest.



## 9 Acknowledgments

This study was supported by the Fond France Canada pour la Recherche (Grant 12/51) to David Au Yang. This project was supported by a grant from the Agence Nationale de la Recherche (ANR) via contract 14-CE33-0009-02-FOFAMIFS. We warmly thank the RSQA members for providing the samples, especially Diane Boulet and Véronique Chalut. We also thank Marjorie Bagur for the logistic help and thank Hao Thi Bui and Boswell Wing for their analyses assistance performed in Montréal and Nelly Assayag and Guillaume Landais for analyses assistance performed at IPGP. IPGP contribution number X.

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
