# Peer review of "Seasonality in the $\Delta^{33}S$ measured in urban aerosols highlights an additional oxidation pathway for atmospheric SO2"

_Atmospheric Chemistry and Physics, 2018_

## Referee Comment (RC1) · H. Bao (Referee) · 8 Nov 2018

General Comments:

The number of reports on modern atmospheric sulfate with non-mass-dependently (NMD) anomalous 33S and/or 36S has been increasing in the last few years. The occurrence of these NMD sulfur isotope anomalies, with $\Delta$33S values ranging from -0.6 to 0.5‰ in modern atmospheric sulfate is puzzling because NMD sulfur isotope signatures were initially assumed to be produced only by high-energy UV photolysis of SO2. In today's atmosphere where O2 is at 21%, only wavelength longer than ~320 nm is available at troposphere where most SO2 emission resides. Thus, SO2

photo-oxidation, instead of SO2 photolysis, has been proposed by some to be a likely mechanism for the observed anomalies. But experimental results do not really match with the sparse observational data. Magnetic isotope effect may play a role in generating the sulfur NMD signatures, but available data do not always have both Δ33S and Δ36S values for checking. A recent paper by Lin et al (PNAS, 2018) best summarized the current state of our knowledge and gaps on the origin and distribution of NMD S isotope anomalies seen among atmospheric sulfate including those volcano and combustion sourced.

At this state of our knowledge, more observational data are badly needed. Although not explicitly expressed or rationalized, Au Yang and his colleagues in this manuscript set out to test the hypothesis that the most positive aerosol sulfate Δ33S value should be found in remote areas far away from the city (Montreal, Canada) and there might be a seasonality change in the Δ33S value due to seasonal contribution change of local anthropogenic emission. They collected PM10 aerosol samples weekly in 2013 from 5 stations in the city of Montreal, Canada and its vicinity. Chemical and multiple S isotope compositions ($\delta$34S, Δ33S, and Δ36S) were measured.

The results reflect some unique aspects of the Montreal PM10 sulfate. For example, the $\delta$34S does not have good seasonality as those observed in Beijing or predicted by some model (Harris et al, 2013). The Δ33S values are largely positive, ranging from -0.08 to 0.34‰ which are similar to values for Beijing's PM10 in summer time (Guo et al., 2010) while very different from the rather negative Δ33S values for Beijing's PM2.5 sulfate in winter (Han et al., 2017). The Montreal PM10 Δ36S data have both positive and negative values and do not have a distinct seasonality.

Au Yang et al then compared their data with existing modern aerosol sulfate data with a focus on the chemical pathways of the atmospheric sulfate formation. The discussion section is very through in coverage. They also proposed their own explanations, albeit rather speculative ones.

Overall, this manuscript offered a much-needed new set of observational data together with suggested new mechanistic interpretations on the puzzling NMD S isotope compositions in modern aerosol sulfate. Although an eventual answer is not given, the data should raise new attention to this persistent puzzle.

Specific Comments:

1. After entertaining various possible mechanisms for the observed NMD S isotope data, the authors settled one of the mechanisms (Page 17, line 4-5): "We suggest that the SO2 photooxidation reaction may occur at the dust surface and, by oxidizing the surrounding SO2 into sulfates, it would deplete the resulting SO4 in 33S and by mass balance, leave the residual SO2 enriched in 33S (Figure 6)." I suggest the authors to keep this proposal short and maybe add that such a hypothesis can be tested in the future via experiments. I think Figure 6 is probably not necessary because it has too many reaction steps and isotope fractionation signs that are themselves very uncertain. The observed sulfate Δ33S data from Antarctica snowpack (Baroni et al., 2007) show that the sulfate Δ33S may change from positive to negative over time during one eruption, suggesting that the SO2 to SO4 conversion step may be associated with a 33S enrichment (Δ33S being positive initially) in product SO4; and it is the leftover SO2 being NMD depleted in 33S which will later turn into SO4. If true, this "elementary" SO2 to SO4 photo-oxidation step in volcanic plumes would have the opposite sign in 33S anomaly to that from tropospheric SO2 oxidation to SO4 as the author proposed. I suggest this difference be discussed.

2. Page 18 line 9-11: Please note that Han et al (2017)'s sulfate were from PM2.5 while Guo et al. (2010) from PM10. The Δ33S values for Han et al are distinctly negative in winter months while for Guo et al's larger particles are distinctly positive in the months of March to August. Therefore, this pattern is not consistent with the authors' prediction of more negative-Δ33S sulfate being preferentially found in larger dust particles. I suggest incorporating this difference in your discussion as well.

Technical corrections:

Page3 line 3-8: $\beta$ value in stable isotope community has been reserved for a fundamental concept, i.e. the equilibrium fractionation factor between a compound and its atomic form of element of interest, e.g., the equilibrium fractionation factor between CO2 and O for oxygen isotopes or SO4 and S for sulfur isotopes (Richet et al., 1977). Most in the triple-isotope community use the Greek symbol $\theta$ to describe the triple sulfur or triple oxygen isotope relationship, such as $33\theta$ and $17\theta$., to avoid confusion. If you insist using $\beta$, please mention $\theta$.

Page3 Line 7 and 11: If the "deviation" at Line 7 refers only to temperature effect, then Line 11 is ok. Otherwise, Line 11's "Non-zero" cases include the deviation mentioned at Line 7. Therefore, either add the term "temperature" at Line 7 or delete "also" at Line 11.

---

## Short Comment (SC1) · 11 Dec 2018

This is an excellent article that provides additional data to the evolving picture of tropospheric sulfate isotope signatures (and the processes that affect them). I would like to see more discussion about the D33S values of the initial SO2 and more justifications / measurements / citations for these assumptions. Iron production in the Quebec / Ontario region of Canada is a high emitter of sulfur dioxide. My first order assumption would be that processing iron from Archean banded iron formations would release SO2 with non mass dependent isotope signatures. SO2 signatures from this region can be transported long distances and may contribute to non mass dependent isotope

signatures significantly downwind (e.g. Boston, MA). It would be useful to understand your reasoning as to why either (1) SO2 emitted from processing iron from banded iron formations will not produce non mass dependent SO2 or (2) this is not a substantial source of SO2 for this region and will not affect the observed isotope signatures. You invoke complex reactions (e.g. SO2 photooxidation and stabilized Criegee intermediates) without constraining the SO2 source signature. A mixing of SO2 from different sources would have different d34S (and likely D33S and D36S) values and may contain seasonality as observed in this study.

---

## Referee Comment (RC2) · Anonymous Referee #2 · 14 Jan 2019

It has been a long question regarding the origin(s) of the non-zero capital delta S-33 and S-36 values observed in tropospheric aerosols. Although earlier studies suggested the potential transfer of stratospheric aerosol could be the source, unknown tropospheric sources or mechanisms can't be ignored. Yang et al reported new capital delta S-33 and S-36 values of sulfate aerosols observed year-round in Montreal, the values show some common features as observed in other locations except for the very low D33S values observed in Beijing. The authors thoughtfully discussed potential sources contributing to the observed non-zero capital delta S-33 and S-36 values, and suggested oxidations by Criegee intermediates and/or photo-oxidation of SO2 on the surface of mineral dust could be responsible for the observations. Although the conclu-

sion is rather speculative, it again calls attentions to the unusual non-zero capital delta S-33 and S-36 values observed frequently in the troposphere, which can neither explained by conventional photo-chemical theme nor well-known oxidation mechanisms.

The proposed photo-oxidation in the presence of mineral dust is inspiring. However, I don't think sulfate with direct or indirect interactions with long-range transported mineral dust would explain the observed D33S patterns, due to the probably small magnitude of this sulfate versus regional or locally produced. The residual SO2 associated with far-away dust source region (from long range transport rooted in Asia or Sahara) maybe just too small to make any difference in D33S measured in Montreal, where local emission of SO2 and the subsequent conversion to sulfate dominate sulfate budget.

In fact, I am confusing by the term of "Photooxidation of SO2 in the presence of mineral dust". as based on the statements in 4.3.4., by Photooxidation of SO2 in the presence of mineral dust, the authors seemed to mean in fact heterogenous SO2 oxidation on the surface of mineral dust or dust enhanced HOx radicals oxidation. If this is the case, then the term of photo-oxidation should be avoided. what's more, if this is the case, the oxidation then should be no difference from that in gaseous and aqueous phase oxidation in terms of the specific oxidation pathways (or oxidant involved), then why large non-zero D33S could be induced?

If it is really photo-oxidation of SO2 that occurs, I don't understand why photo-oxidation only could occur on the surface of mineral dust. If mineral dust serves as the reaction site promoting the photo-oxidation of SO2, why not other aerosols? There are studies indicating the photolysis rate of nitrate on aerosols a few orders of magnitude larger than that in the gas phase, could be this the case of SO2? What's more, lab experiments, model calculations and ice-core data indicated when photooxidation occurs, the formed sulfate is in general enriched in S-33 and the residual SO2 is depleted in S-33, why in this case assuming the opposite pattern? Or just to fulfill the observation?

My last comments are about the difference between the Beijing and Montreal samples.

There are few things might be important but the authors seemed to not pay enough attentions. 1. The different SO2 source in the two cities, especially in winter, heating source should be the main source of SO2 but what is the difference of the energy structure between the two cities? 2. The aerosols being collected and measured, one is PM2.5 (the fine mode) and the other is PM10 (the coarse mode). The high Na+ concentration in Montreal also indicate the difference, as sea-salt aerosols are often in the coarse mode. Would be sulfate formed in or associated with coarse mode aerosols isotopically different with that in fine mode? It could be another way around, as photo-oxidation of SO2 with coarse mode aerosols leads to sulfate enriched in S-33, leaving residual SO2 depleted in S-33 and which is ultimately converted to sulfate by heterogenous reaction in fine mode aerosols or by gaseous oxidation and then nucleate to or scavenged by fine mode aerosols. Just brainstorming as no concrete answer based on current knowledge available.

---

## Author Comment (AC1) · 25 Feb 2019

Answer to review for MS acp-2018-1091

Au Yang et al., Seasonality in the $\Delta$33S measured in urban aerosols highlights an additional oxidation pathway for atmospheric SO2 https://doi.org/10.5194/acp-2018-1091

We thank the two referees and Dr. Whitehill for evaluating our manuscript and providing us with feedbacks on its scientific content. The reviewers did not express any significant rebuttals, agreeing in particular with our proposition that the $\Delta$33S-anomalies are transported to rather than produced downtown of the city. Most of the

comments/suggestions were made to clarify/precise the SO2 photooxidation process in presence of mineral dust: we included their comments, which, we believe, greatly improve the clarity of our manuscript. In particular:

- We modified section 4.4.4 Photo-oxidation where 1) we added a discussion on the distinct isotope behavior between SO2 oxidation on mineral dust (our model) and in the stratosphere, as recorded in the Antarctica snowpack 2) We modified the discussion when comparing $\Delta 33S$ signal between Montréal and Beijing (taking into account the different in energy sources present in both cities) and discuss differences between PM2.5 and PM10, which was not present in the original version of our manuscript.

- We added a discussion in section 4.1 "Anthropogenic emissions, $\Delta 33S$-values and seasonality" the origin of the SO2 and whether emitted SO2 could have non-zero $\Delta 33S$ due to the source-material having non-zero $\Delta 33S$" following the point expressed by Dr. Whitehill and R#2. Reviewers' comments appear below in blue; our detailed answers are in black and related modifications in the manuscript are reported in italics.

Referee #1: Dr. Huiming Bao

General comments : 1. After entertaining various possible mechanisms for the observed NMD S isotope data, the authors settled one of the mechanisms (Page 17, line 4-5): "We suggest that the SO2 photooxidation reaction may occur at the dust surface and, by oxidizing the surrounding SO2 into sulfates, it would deplete the resulting SO4 in 33S and by mass balance, leave the residual SO2 enriched in 33S (Figure 6)." I suggest the authors to keep this proposal short and maybe add that such a hypothesis can be tested in the future via experiments.

We agree with the reviewer's point, that this hypothesis may not be the sole explanation to the positive and negative $\Delta 33S$-values measured in aerosols. This hypothesis, as it is described, remains speculative and could be one of the many others oxidation pathways to consider. We find no objection to keep the text concise, but still need to explain/strengthen our point and develop possible ways to test our model (R#2 was

confused with our wording/explanation). We have clarified the text accordingly l.4 p.18.

I think Figure 6 is probably not necessary because it has too many reaction steps and isotope fractionation signs that are themselves very uncertain.

We understand the reviewer's point. However, we believe that Figure 6 is useful for the reader as it is meant to clarify our view and feel that it should be kept as is. Still, given that these reactions are described in the text, we simplified to our best Figure 6 in the new version of the manuscript. The observed sulfate $\Delta$33S data from Antarctica snowpack (Baroni et al., 2007) show that the sulfate $\Delta$33S may change from positive to negative over time during one eruption, suggesting that the SO2 to SO4 conversion step may be associated with a 33S enrichment ($\Delta$33S being positive initially) in product SO4; and it is the leftover SO2 being NMD depleted in 33S which will later turn into SO4. If true, this "elementary" SO2 to SO4 photo-oxidation step in volcanic plumes would have the opposite sign in 33S anomaly to that from tropospheric SO2 oxidation to SO4 as the author proposed. I suggest this difference be discussed.

This is a good point that was worth being mentioned, a point also highlighted by R#2 (see below). → A section l.20 p.18 was added: " It is worth mentioning that our model would thus generate a different temporal pattern from the one recorded in sulfates from the Antarctica snowpack which are first characterized by positive $\Delta$33S-values that then shift to negative $\Delta$33S-values, reflecting a depletion in 33S in the residual SO2 pool (Baroni et al., 2008;Gautier et al., 2018). Although the origin of the $\Delta$33S-values in snowpack remains unclear, a combination of different oxidation pathways with similar contributions of S-MDF (high or lower contribution of OH oxidation pathway) and S-MIF processes (photoexcitation and photolysis) has been recently suggested to explain such $\Delta$33S-values (Gautier et al., 2018). The OH oxidation pathway is occurring in both the troposphere and the stratosphere. However, in the troposphere as i) photolysis cannot occur because of the ozone layer and ii) photooxidation would only occur in a narrow range of UV (see section 4.4.2.) but would unlikely display a seasonal variation, we suggest that the reactions responsible for S-MIF in the stratosphere and in the

troposphere are different. Thus, the contrasting patterns observed in sulfates in Antarctica and in Montreal could be explained by the implication of different combinations of oxidation pathways where a S-MIF process other than photolysis and photooxidation is involved."

2. Page 18 line 9-11: Please note that Han et al (2017)'s sulfate were from PM2.5 while Guo et al. (2010) from PM10. The $\Delta^{33}S$ values for Han et al are distinctly negative in winter months while for Guo et al's larger particles are distinctly positive in the months of March to August. Therefore, this pattern is not consistent with the authors' prediction of more negative-$\Delta^{33}S$ sulfate being preferentially found in larger dust particles. I suggest incorporating this difference in your discussion as well.

We agree with the reviewer that including the study of Guo et al. and highlighting the difference in aerosols sizes between Guo et al and Han et al. (PM10 versus PM2.5) in the discussion is useful. Guo et al.' was not discussed in our original manuscript as the authors did not report chemical analysis for their aerosols, making it impossible to determine whether the PM10 could record or be characterized by a higher contribution of dust than PM2.5. This is now discussed in the revised version of the manuscript. → A section l.33 p.19 was added: "Negative $\Delta^{33}S$ values have also not been measured in PM10 during spring. Guo et al. (2010) data show positive $\Delta^{33}S$-values, similar to ours and to other studies but different from Han et al. (2017). However, Guo et al. (2010) did not report major elements in their aerosol samples, making it difficult to detect any significant dust contribution. Nevertheless while Guo et al. (2010) measured sulfates S isotope compositions until April 11th, Cao et al. (2014) reported a significant dust event on April 27th of the same year. In that respect this does not contradict our hypothesis: SO2 photooxidation on mineral dust could lead to positive $\Delta^{33}S$ of the residual SO2 transported to Beijing. Moreover, for our model to be consistent with the data of Han et al. (2017), their aerosol fine fraction would need to be dominated by dust which is consistent with the observation that Asian dust storms contribute to the PM2.5 budget in Beijing (Han et al., 2015)."

Technical corrections: Page3 line 3-8: $\beta$ value in stable isotope community has been reserved for a fundamental concept, i.e. the equilibrium fractionation factor between a compound and its atomic form of element of interest, e.g., the equilibrium fractionation factor between CO2 and O for oxygen isotopes or SO4 and S for sulfur isotopes (Richet et al., 1977). Most in the triple-isotope community use the Greek symbol $\theta$ to describe the triple sulfur or triple oxygen isotope relationship, such as $33\theta$ and $17\theta$., to avoid confusion. If you insist using $\beta$, please mention $\theta$.

This is not exactly true: $\beta$-factor is indeed used to express the reduced partition function of molecules from which equilibrium fractionation factors are calculated. However, $\beta$-values are also used to express mass laws between two isotopic systems in general (e.g. Young and Galy, 2004;) while $\theta$- is specifically used when it corresponds to isotope equilibrium (e.g. Farquhar and Wing, 2003; Dauphas and Schauble, 2016),  is used to define the slope defined by data which, owing to mass conservation effects, does not necessarily correspond to $\beta$- (i.e. the mass exponent relating the two isotope fractionation factors (Farquhar and Wing, 2003, Ono et al., 2006; Johnston et al., 2008). Given the remaining lack of understanding on the reactions involved and the mechanisms (equilibrium, kinetic, etc. . .), we feel that we cannot use the $\theta$-notation. Using $\beta$- is, we think, therefore more appropriate. We have added some clarifications to help the reader with the 'isotope notations' which, we agree, can easily be confusing. → We modified l.5, p3: " The $\beta$-exponent is usually expressed as  to refer to isotope equilibrium. We are using $\beta$- instead as the processes describing the SO2-oxidation are actually not at the isotope equilibrium. Its value depends on the reaction considered (Farquhar et al., 2001;Harris et al., 2013;Ono et al., 2013;Watanabe et al., 2009). At high temperature (> 500°C, i.e. under equilibrium), 33 and 36-values are respectively 0.515 and 1.889 (Eldridge et al., 2016;Otake et al., 2008)"

Page3 Line 7 and 11: If the "deviation" at Line 7 refers only to temperature effect, then Line 11 is ok. Otherwise, Line 11's "Non-zero" cases include the deviation mentioned at Line 7. Therefore, either add the term "temperature" at Line 7 or delete "also" at Line

11. → This has been modified in the text.

Anonymous Referee #2 I don't think sulfate with direct or indirect interactions with long-range transported mineral dust would explain the observed $\Delta33S$ patterns, due to the probably small magnitude of this sulfate versus regional or locally produced. The residual $SO_2$ associated with far-away dust source region (from long range transport rooted in Asia or Sahara) maybe just too small to make any difference in $\Delta33S$ measured in Montreal, where local emission of $SO_2$ and the subsequent conversion to sulfate dominate sulfate budget.

This is a good point raised by the reviewer. However it would be possible to account for the $\Delta33S$-values with a small contribution of this oxidation pathway. If 1) the sulfates formed by photooxidation on mineral dust were characterized by high $\Delta33S$-values (hypothetically 10‰, and 2) would hypothetically contribute to $\sim 10\%$ of the total sulfate, then even a small contribution of those sulfates, mixed with sulfates formed by the major oxidation pathways, which are locally produced (i.e. $\Delta33S \sim 0‰$, could explain the $\Delta33S$ -values observed in the troposphere ($\Delta33S < 0.5$). → We have made this point clearer l.15 p.19 : "If this latest oxidation pathway could promote the formation of sulfates characterized by high $\Delta33S$-values (hypothetically 10‰, then a small contribution (hypothetically $\sim10\%$) from this oxidation pathway would produce a significant signal on the sulfur isotope composition of tropospheric sulfate aerosols (i.e. $\Delta33S \sim 1‰$ based on these hypotheses). In this case, even a small proportion of those sulfates mixed with sulfates formed by the major oxidation pathways locally produced (i.e. $\Delta33S \sim 0‰$ could explain the $\Delta33S$-values observed in the troposphere ($\Delta33S <0.5‰$. This hypothesis needs to be further tested."

In fact, I am confusing by the term of "Photooxidation of $SO_2$ in the presence of mineral dust". as based on the statements in 4.3.4., by Photooxidation of $SO_2$ in the presence of mineral dust, the authors seemed to mean in fact heterogenous $SO_2$ oxidation on the surface of mineral dust or dust enhanced HOx radicals oxidation. If this is the case, then the term of photo-oxidation should be avoided. Our use of 'photooxidation'

is actually meant to be consistent with the literature reporting "photooxidation of SO2 on mineral dust" (Yu et al., 2017;He et al., 2014;Dupart et al., 2012;George et al., 2015;Usher et al., 2003;Zhao et al., 2018;Ma et al., 2018).

what's more, if this is the case, the oxidation then should be no difference from that in gaseous and aqueous phase oxidation in terms of the specific oxidation pathways (or oxidant involved), then why large non-zero D33S could be induced? We agree about the specific mechanism (SO2 oxidation by OH) but little is known about the number of other mechanisms behind 'the in-particle chemistry'; heterogeneous SO2 oxidation by OH radicals have been described (as mentioned by R#2) but oxidation by other radicals may exist, such as the recently identified superoxide O2Åů. The S-isotope fractionations associated with this latest oxidation pathway have not been reported yet. Furthermore, others radicals/oxidants might as well be discovered in the future, leading SO2 oxidation by mineral dust to produce specific S-isotope fractionation factors compared to OH-oxidation. → We included R#2's comment and have clarified our point accordingly: "To date, the mechanisms behind the in-particle chemistry remain little studied and several SO2 heterogeneous oxidation reactions may have been overlooked" l.7 p.16. We also changed the text as follows: "The oxidation implicating heterogeneous oxidation and OH radicals should a priori not show significant differences from the one that occurs in gaseous and aqueous phase; i.e. heterogeneous oxidation of SO2 is likely to induce a mass dependent fractionation of S-isotopes (Harris et al., 2012) with resulting negative $\Delta$33S-values <-0.15‰ (Harris et al., 2013). Among other reactions, SO2 oxidation by the O2Åů superoxide radical anion is another oxidation reaction that has not yet been isotopically characterized (Dupart et al., 2014;Usher et al., 2003). If this latest oxidation pathway could promote the formation of sulfates characterized by high $\Delta$33S-values (hypothetically 10‰, then a small contribution (hypothetically ∼10%) from this oxidation pathway would produce a significant signal on the sulfur isotope composition of tropospheric sulfate aerosols (i.e. $\Delta$33S ∼ 1‰ based on these hypotheses). In this case, even a small proportion of those sulfates mixed with sulfates formed by the major oxidation pathways locally produced (i.e. $\Delta$33S ∼

0‰ could explain the $\Delta^{33}$S-values observed in the troposphere ($\Delta^{33}$S <0.5‰. This hypothesis needs to be further tested " l.10 p.19

If it is really photo-oxidation of SO2 that occurs, I don't understand why photo-oxidation only could occur on the surface of mineral dust. If mineral dust serves as the reaction site promoting the photo-oxidation of SO2, why not other aerosols? It may well be but, to our knowledge, no study has ever reported photooxidation of SO2 in the presence of other types of C-rich, i.e. 'organic,' aerosols. This could result from the low concentration of metal oxides in other aerosols (Cass et al., 2000).

There are studies indicating the photolysis rate of nitrate on aerosols a few orders of magnitude larger than that in the gas phase, could be this the case of SO2 ? Compared to nitrate, photolysis of the sulfate is less likely because the main wavelength region of SO2-absorption (190-220 nm) is filtered by the ozone layer. However, as stated in section 4.3.2, a narrow wavelength range (typically 320 to 330 nm) where SO2 absorption could occur (i.e. not filtered by the ozone layer) in the troposphere, which still leaves room for S-MIF to be produced. This is better stated in the text l.23 p.15.

What's more, lab experiments, model calculations and ice-core data indicated when photooxidation occurs, the formed sulfate is in general enriched in S-33 and the residual SO2 is depleted in S-33 We have addressed this point above.

Why in this case assuming the opposite pattern? Or just to fulfill the observation? Indeed, this is inferred, not observed. The text now states more clearly this aspect l.4 p18

1. The different SO2 source in the two cities, especially in winter, heating source should be the main source of SO2 but what is the difference of the energy structure between the two cities?

R#2's comment can be understood in two ways: (a) distinct oxidation processes of sulfur dioxide with $\Delta^{33}$S ∼ 0 ‰ (here low vs high temperature of combustion) leading to

markedly distinct isotope signals in Beijing aerosols and (b) distinct sulfur sources having distinct, non-zero $\Delta 33S$. Point 'b' is somewhat similar to the comment expressed below by Dr Whitehill. a- Both Montréal and Beijing have their energy relying primarily on coal and oil burning. The main difference probably lies in the temperature of combustion of coal and wood (see Han et al., 2016; Lin et al. 2018) where 'low temperature combustion' would be more significant in Beijing and considered as a possibility to account for the distinctly negative $\Delta 33S$ of aerosols (Han et al., 2016; Lin et al., 2017). However, Lin et al. (2018) recently questioned this mechanism on the basis of new results (low temperature combustion would not lead to very anomalous $\Delta 33S$, yet with still significant non-zero $\Delta 36S$-values). We cannot discard this possibility but tried, using available data, to investigate whether SO2 photo-oxidation on mineral dust could represent a viable alternative hypothesis. The revised manuscript states more clearly that additional data are required to discuss such a possibility. b- As suggested by Dr Whitehill in his comment, the question could also relate to distinct sources of sulfur with different $\Delta 33S$ -values. This is addressed below.

→We modified the discussion l.30 p.12: " This contrasts with the interpretation where the negative $\Delta 33S$-values (down to -0.6‰ measured during winter in Beijing would relate to anthropogenic sources, in particular those generating incomplete, i.e. low-temperature, coal or wood combustion (Han et al., 2017). Still, this model cannot explain the total range of isotope compositions observed. The authors mostly rely on data showing that primary aerosols are characterized by negative $\Delta 33S$-values but only down to -0.2‰ (Lee et al., 2002). Also the complementary positive $\Delta 33S$ still need to be addressed. Furthermore, Han et al. (2017) interpretation would predict: i) a seasonality with negative $\Delta 33S$-values down to -0.6‰ during winter as a result from increased coal and wood burning and ii) a gradient in the $\Delta 33S$-values from the outer towards the inner city with isotope shifting from $\sim$ 0‰ to negative $\Delta 33S$-values. This would contradict our observations, since our data in Montreal show the opposite to what was observed in Beijing. It comes that based on the available data of S anthropogenic emissions, the combustion of coal or wood at low temperature can neither explain

Interactive
comment

the ∆33S seasonality nor the highest ∆33S-values up to 0.5‰ measured in urban aerosols. This conclusion is reinforced by the fact that coal is not the major source of energy in Montreal, oil representing 50% of the fuel energy in Quebec (Montréal, 2015). Oil would thus display ∆33S-values close to 0‰ (Lee et al., 2002). Taken together, our observations suggest that anthropogenic activities (both coal and oil combustion) are unlikely responsible for the ∆33S seasonality nor the highest ∆33S-values up to 0.5‰ measured in Montreal urban aerosols. This implies that non-zero ∆33S-values are produced in rural rather than in urban environments. Thus, the following discussion mostly focuses on data from station 98, located on the western part of the island, upstream of the main blowing winds and supposedly less affected by emissions from local anthropogenic activities.".

2. The aerosols being collected and measured, one is PM2.5 (the fine mode) and the other is PM10 (the coarse mode). The high Na+ concentration in Montreal also indicates the difference, as sea-salt aerosols are often in the coarse mode. Would be sulfate formed in or associated with coarse mode aerosols isotopically different with that in fine mode? It could be another way around, as photo-oxidation of SO2 with coarse mode aerosols leads to sulfate enriched in S33, leaving residual SO2 depleted in S-33 and which is ultimately converted to sulfate by heterogeneous reaction in fine mode aerosols or by gaseous oxidation and then nucleate to or scavenged by fine mode aerosols. Just brainstorming as no concrete answer based on current knowledge available.

The reviewer raises an interesting issue and we agree that S-isotopes of sulfates formed in the aerosol coarse mode could be different from the one in the fine mode fraction. To our knowledge, no study (neither experimental nor with natural samples) has ever been published yet but we fully agree that this is a prediction that can be made from our model, which we have included in the revised manuscript. Besides, the hypothesis expressed by the reviewer: "It could be another way around,as photo-oxidation of SO2 with coarse mode aerosols leads to sulfate enriched in S33, leaving

residual SO2 depleted in S-33 and which is ultimately converted to sulfate by heterogenous reaction in fine mode aerosols or by gaseous oxidation and then nucleate to or scavenged by fine mode aerosols" is also interesting. However, following this hypothesis, positive Δ33S-values would have been explained by the input of sulfates associated with dust, which is not consistent with our chemical analyses that do not indicate any contributions of mineral dust to Montreal aerosols. Moreover, in this case, we would expect negative Δ33S-values on sulfates collected at station 98 (the station least impacted by anthropogenic emissions) which is not the case. For these reasons, this hypothesis cannot account for our observations. We made this clearer in the manuscript.

→ We modified the discussion l.4 p.18 as follows: "Thus, in order to explain our data (i.e. most positive Δ33S-values at station 98, seasonality of the S-isotope compositions, no dust particles detected in Montreal aerosols), we suggest that the SO2 photooxidation reaction may occur at the dust surface and, by oxidizing the surrounding SO2 into sulfates, it would deplete the resulting SO4 in 33S and by mass balance, leave the residual SO2 enriched in 33S (Figure 6). Sulfates associated to dust would be characterized by negative Δ33S-values and will be deposited while the residual atmospheric SO2 (i.e. characterized by positive Δ33S-values) would be transported to Montreal. The transported SO2 enriched in 33S would then be oxidized into sulfates in Montreal vicinity through the major oxidation pathways (O2+TMI, H2O2, O3, OH, NO2). We suggest the presence of two different types of sulfates: i) the first type would be formed by photooxidation and would be associated to coarse particles (dust particles) while ii) the second type would be formed by the oxidation of the remaining SO2 and thus likely be associated to finer particles. These sulfates supposedly characterized by positive Δ33S up to 0.5‰ would be mixed with both primary and secondary sulfates emitted and formed within the city and supposedly characterized by Δ33S-values close to 0‰ (i.e. oxidation by O2+TMI, H2O2, O3, OH, NO2 ; Figure 6). " "

Comment #1 : Dr. Andrew Whitehill I would like to see more discussion about the

$\Delta 33S$ values of the initial $SO_2$ and more justifications / measurements / citations for these assumptions. Iron production in the Quebec / Ontario region of Canada is a high emitter of sulfur dioxide. My first order assumption would be that processing iron from Archean banded iron formations would release $SO_2$ with non mass dependent isotope signatures. $SO_2$ signatures from this region can be transported long distances and may contribute to non mass dependent isotope signatures significantly downwind (e.g. Boston, MA). It would be useful to understand your reasoning as to why either (1) $SO_2$ emitted from processing iron from banded iron formations will not produce non mass dependent $SO_2$ or (2) this is not a substantial source of $SO_2$ for this region and will not affect the observed isotope signatures. You invoke complex reactions (e.g. $SO_2$ photooxidation and stabilized Criegee intermediates) without constraining the $SO_2$ source signature. A mixing of $SO_2$ from different sources would have different $\delta 34S$ (and likely $\Delta 33S$ and $\Delta 36S$) values and may contain seasonality as observed in this study.

Below are the answers to the two questions. It would be useful to understand your reasoning as to why (2) this is not a substantial source of $SO_2$ for this region and will not affect the observed isotope signatures

The large majority of coal and oil used worldwide for energy are derived from Proterozoic sediments (<2.3 Gy) and, as such, does not have significant non-zero $\Delta 33S$ (typically within $\pm 0.1‰$ e.g. Farquhar and Wing, 2003). The complete conversion of sulfur (as organic S, sulfate and/or pyrite) to $SO_2$ implies that it would have the same isotope composition than that of its starting material, i.e. no isotope fraction and $\Delta 33S = 0.0 \pm 0.1‰$. Only if part of the $SO_2$ is scavenged and the fractionation process is strongly non-mass dependent (beta $\neq$ 0.515) would the emitted $SO_2$ have non-zero $\Delta 33S$. 'Low temperature combustion' was suggested to represent such a process. However, this is dealing with identifying processes. Dr Whithill and R#2 wonders whether the source of sulfur could be characterized by non-zero $\Delta 33S$, with an emphasis on iron mining, which relies on the extraction of some Archean banded-iron formation mining. These

kinds of samples can have high S-content (several percent) with significant non-zero Δ33S (typically positive Δ33S-values, up to several per mille). This is a possibility that we did not originally address.

Iron production in the Quebec / Ontario region of Canada is indeed an emitter of sulfur dioxide. However, with a total of 5800 tons of SO2 emission each year (Environnement Canada, 2013), this only represents 1.5% of the total 370000 tons of SO2 emitted (Environnement Canada, 2013). It would be useful to understand your reasoning as to why (1) SO2 emitted from processing iron from banded iron formations will not produce non mass dependent SO2 By considering that 1) the BIF are characterized by a Δ33S-values up to 2‰ (Thomassot et al., 2015), 2) iron process does not fractionate the sulfur isotopes (similar to coal combustion) and 3) only 1.5% of the SO2 results from iron processing (Environnement Canada, 2013), this would only produce Δ33S-anomalies up to 0.02‰Thus, iron processing can hardy account for the origin of non-zero Δ33S-values observed in most aerosols. More specifically, Canadian iron ore production is split in ∼50-45% between Quebec and Labrador (with 5% in Nunavut). With respect to Quebec, iron production is mainly from Algoma BIFs of about 2.8-2.7 Ga, typified by the Temagami deposit for which there are recently available 33S data (Diekrup et al., 2018). From their Table 1, all their sedimentary data (oxidic facies, cherts, BIF sulphides; sulphidic clays; sulphide veins) gives an average of 0.467 +/- 0.707 (one standard deviation; n=50), which makes emitted SO2 having even smaller Δ33S.

→ Section 4.1 " Anthropogenic emission, Δ33S -values and seasonality" has been modified "l.3 p.12 : "The large majority of coal and oil used worldwide as an energy source are extracted from Proterozoic sediments (<2.3 Gy) and, as such, does not have significant non-zero Δ33S (typically within ±0.1‰ e.g. Farquhar and Wing, 2003). The complete conversion of sulfur (as organic S, sulfate and/or pyrite) to SO2 implies that SO2 has the same isotope composition than that of its starting material, i.e. no isotope fractionation or Δ33S = 0.0±0.1‰Only if part of the SO2 is scavenged

and the fractionation process is strongly non-mass dependent ($\beta \neq 0.515$) would the emitted SO2 have a non-zero $\Delta$33S. Han et al. (2016) suggested that the combustion of coal or wood at low temperature may represent such conditions. Iron extraction from Archean banded-iron formation (BIF) is another source of atmospheric S that therefore produces non-zero $\Delta$33S. However, the 5,800 tons of SO2 emitted each year in the Quebec/Ontario region by mining activities (Environnement Canada, 2013) only represents 1.5% of the annual 370,000 tons of national SO2 emissions (Environnement Canada, 2013). If we consider a high average $\Delta$33S of 2‰ (Thomassot et al., 2015) and a proportion of 1.5% of SO2 resulting from iron processing, this would lead to an average $\Delta$33S-anomaly of the final SO2 of up to 0.02‰Thus, iron processing can hardy account for the origin of the non-zero $\Delta$33S-values observed in most aerosols. More specifically, the Canadian iron ore production is split between Quebec (50%), Labrador (45%) and Nunavut (5%). With respect to Quebec, iron production is mainly operated from the Algoma BIFs ($\sim$2.8 Ga) typified by the Temagami deposits for which Diekrup et al. (2018) give an average $\Delta$33S of 0.467$\pm$0.707 (samples including oxidic facies, cherts, BIF sulphides and sulphidic clays, sulphide veins), which makes the emitted SO2 having even smaller $\Delta$33S. In the following discussion we will therefore consider that sources of SO2 have $\Delta$33S$\sim$0‰ and that only specific chemical reactions (photochemical or not) can produce non-zero $\Delta$33S."

Reference

Boulet, D., and S. Melançon (2016), Bilan environnemental. Qualité de l'air à Montréal, Rapport Annuel 2016. Ville de Montréal, Service de l'environnement Division de la planification et du suivi environnemental, RSQA, 12. Lee, C. W., J. Savarino, H. Cachier, and M. Thiemens (2002), Sulfur (32S, 33S, 34S, 36S) and oxygen (16O, 17O, 18O) isotopic ratios of primary sulfate produced from combustion processes, Tellus B, 54(3), 193-200. Lin, M., X. Zhang, M. Li, Y. Xu, Z. Zhang, J. Tao, B. Su, L. Liu, Y. Shen, and M. H. Thiemens (2018), Five-S-isotope evidence of two distinct mass-independent sulfur isotope effects and implications for the modern and Archean atmospheres, Proceedings of the National Academy of Sciences, 115(34), 8541-8546. Thomassot, E., J. O'Neil, D. Francis, P. Cartigny, and B. A. Wing (2015), Atmospheric record in the Hadean Eon from multiple sulfur isotope measurements in Nuvvuagittuq Greenstone Belt (Nunavik, Quebec), Proceedings of the National Academy of Sciences, 112(3), 707-712. Baroni, M., Savarino, J., Cole‐Dai, J., Rai, V. K., and Thiemens, M. H.: Anomalous sulfur isotope compositions of volcanic sulfate over the last millennium in Antarctic ice cores, Journal of Geophysical Research: Atmospheres, 113, 2008. Cass, G. R., Hughes, L. A., Bhave, P., Kleeman, M. J., Allen, J. O., and Salmon, L. G.: The chemical composition of atmospheric ultrafine particles, Philosophical Transactions of the Royal Society of London A: Mathematical, Physical and Engineering Sciences, 358, 2581-2592, 2000. Diekrup, D., Hannington, M. D., Strauss, H., and Ginley, S. J.: Decoupling of Neoarchean sulfur sources recorded in Algoma-type banded iron formation, Earth and Planetary Science Letters, 489, 1-7, 2018. Dupart, Y., King, S. M., Nekat, B., Nowak, A., Wiedensohler, A., Herrmann, H., David, G., Thomas, B., Miffre, A., and Rairoux, P.: Mineral dust photochemistry induces nucleation events in the presence of SO2, Proceedings of the National Academy of Sciences, 109, 20842-20847, 2012. Dupart, Y., Fine, L., D'Anna, B., and George, C.: Heterogeneous uptake of NO2 on Arizona Test Dust under UV-A irradiation: An aerosol flow tube study, Aeolian Research, 15, 45-51, 2014. Eldridge, D., Guo, W., and Farquhar, J.: Theoretical estimates of equilibrium sulfur isotope effects in aqueous sulfur systems: Highlighting the role of isomers in the sulfite and sulfoxylate systems, Geochimica et Cosmochimica Acta, 195, 171-200, 2016. Environnement Canada: National Pollutant Release Inventory, 2013. Farquhar, J., Savarino, J., Airieau, S., and Thiemens, M. H.: Observation of wavelength‐sensitive mass‐independent sulfur isotope effects during SO2 photolysis: Implications for the early atmosphere, Journal of Geophysical Research: Planets (1991–2012), 106, 32829-32839, 2001. Gautier, E., Savarino, J., Erbland, J., and Farquhar, J.: SO2 oxidation kinetics leave a consistent isotopic imprint on volcanic ice core sulfate, Journal of Geophysical Research: Atmospheres, 2018. George, C., Ammann, M., D'Anna, B., Donaldson, D., and Nizkorodov, S. A.: Heterogeneous photochemistry in the atmosphere, Chemical reviews, 115, 4218-4258, 2015. Han, L., Cheng, S., Zhuang, G., Ning, H., Wang, H., Wei, W., and Zhao, X.: The changes and long-range transport of PM2. 5 in Beijing in the past decade, Atmospheric Environment, 110, 186-195, 2015. Han, X., Guo, Q., Strauss, H., Liu, C.-Q., Hu, J., Guo, Z., Wei, R., Peters, M., Tian, L., and Kong, J.: Multiple sulfur isotope constraints on sources and formation processes of sulfate in Beijing PM2. 5 aerosol, Environmental Science & Technology, 2017. Harris, E., Sinha, B., Foley, S., Crowley, J., Borrmann, S., and Hoppe, P.: Sulfur isotope fractionation during heterogeneous oxidation of SO 2 on mineral dust, Atmospheric Chemistry and Physics, 12, 4867-4884, 2012. Harris, E., Sinha, B., Hoppe, P., and Ono, S.: High-precision measurements of 33S and 34S fractionation during SO2 oxidation reveal causes of seasonality in SO2 and sulfate isotopic composition, Environmental science & technology, 47, 12174-12183, 2013. He, H., Wang, Y., Ma, Q., Ma, J., Chu, B., Ji, D., Tang, G., Liu, C., Zhang, H., and Hao, J.: Mineral dust and NOx promote the conversion of SO2 to sulfate in heavy pollution days, Scientific reports, 4, 2014. Lee, C. W., Savarino, J., Cachier, H., and Thiemens, M.: Sulfur (32S, 33S, 34S, 36S) and oxygen (16O, 17O, 18O) isotopic ratios of primary sulfate produced from combustion processes, Tellus B, 54, 193-200, 2002. Ma, J., Chu, B., Liu, J., Liu, Y., Zhang, H., and He, H.: NO x promotion of SO 2 conversion to sulfate: An important mechanism for the occurrence of heavy haze during winter in Beijing, Environmental Pollution, 233, 662-669, 2018. Montréal, V. d.: Reduced dependance on fossil fuels in Montréal 2015. Ono, S., Whitehill, A., and Lyons, J.: Contribution of isotopologue self‐shielding to sulfur mass‐independent fractionation during sulfur dioxide photolysis, Journal of Geophysical Research: Atmospheres, 118, 2444-2454, 2013. Otake, T., Lasaga, A. C., and Ohmoto, H.: Ab initio calculations for equilibrium fractionations in multiple sulfur isotope systems, Chemical Geology, 249, 357-376, 2008. Thomassot, E., O'Neil, J., Francis, D., Cartigny, P., and Wing, B. A.: Atmospheric record in the Hadean Eon from multiple sulfur isotope measurements in Nuvvuagittuq Greenstone Belt (Nunavik, Quebec), Proceedings of the National Academy of Sciences, 112, 707-712, 2015. Usher, C. R., Michel, A. E., and Grassian, V. H.: Reactions on mineral dust, Chemical Reviews, 103, 4883-4940, 2003. Watanabe, Y., Farquhar, J., and Ohmoto, H.: Anomalous fractionations of sulfur isotopes during thermochemical sulfate reduction, Science, 324, 370-373, 2009. Yu, Z., Jang, M., and Park, J.: Modeling atmospheric mineral aerosol chemistry to predict heterogeneous photooxidation of SO 2, Atmospheric Chemistry and Physics, 17, 10001-10017, 2017. Zhao, D., Song, X., Zhu, T., Zhang, Z., Liu, Y., and Shang, J.: Multiphase oxidation of SO 2 by NO 2 on CaCO 3 particles, Atmospheric Chemistry and Physics, 18, 2481-2493, 2018.

---

## Author Response (AR1)

**Answer to review for MS acp-2018-1091**

Au Yang et al., Seasonality in the $\Delta^{33}$S measured in urban aerosols highlights an additional oxidation pathway for atmospheric $SO_2$ **https://doi.org/10.5194/acp-2018-1091**

We thank the two referees and Dr. Whitehill for evaluating our manuscript and providing us with feedbacks on its scientific
5   content. The reviewers did not express any significant rebuttals, agreeing in particular with our proposition that the $\Delta^{33}$S-anomalies are transported to rather than produced downtown of the city. Most of the comments/suggestions were made to clarify/precise the $SO_2$ photooxidation process in presence of mineral dust: we included their comments, which, we believe, greatly improve the clarity of our manuscript. In particular:

- We modified section 4.4.4 Photo-oxidation where 1) we added a discussion on the distinct isotope behavior between $SO_2$
10   oxidation on mineral dust (our model) and in the stratosphere, as recorded in the Antarctica snowpack 2) We modified the discussion when comparing $\Delta^{33}$S signal between Montréal and Beijing (taking into account the different in energy sources present in both cities) and discuss differences between $PM_{2.5}$ and $PM_{10}$, which was not present in the original version of our manuscript.

- We added a discussion in section 4.1 "Anthropogenic emissions, $\Delta^{33}$S-values and seasonality" the origin of the $SO_2$ and
15   whether emitted $SO_2$ could have non-zero $\Delta^{33}$S due to the source-material having non-zero $\Delta^{33}$S" following the point expressed by Dr. Whitehill and R#2.

Reviewers' comments appear below in blue; our detailed answers are in black and related modifications in the manuscript are reported in italics.

**Referee #1: Dr. Huiming Bao**

20   General comments :

1. After entertaining various possible mechanisms for the observed NMD S isotope data, the authors settled one of the mechanisms (Page 17, line 4-5): "We suggest that the SO2 photooxidation reaction may occur at the dust surface and, by oxidizing the surrounding SO2 into sulfates, it would deplete the resulting SO4 in 33S and by mass balance, leave the residual SO2 enriched in 33S (Figure 6)." I suggest the authors to keep this proposal short and maybe add that such a hypothesis can
25   be tested in the future via experiments.

We agree with the reviewer's point, that this hypothesis may not be the sole explanation to the positive and negative $\Delta^{33}$S-values measured in aerosols. This hypothesis, as it is described, remains speculative and could be one of the many others oxidation pathways to consider. We find no objection to keep the text concise, but still need to explain/strengthen our point and develop possible ways to test our model (R#2 was confused with our wording/explanation). We have clarified the text
30   accordingly l.4 p.18.

I think Figure 6 is probably not necessary because it has too many reaction steps and isotope fractionation signs that are themselves very uncertain.

We understand the reviewer's point. However, we believe that Figure 6 is useful for the reader as it is meant to clarify our view and feel that it should be kept as is. Still, given that these reactions are described in the text, we simplified to our best Figure 6 in the new version of the manuscript.

The observed sulfate $\Delta^{33}S$ data from Antarctica snowpack (Baroni et al., 2007) show that the sulfate $\Delta^{33}S$ may change from positive to negative over time during one eruption, suggesting that the $SO_2$ to $SO_4$ conversion step may be associated with a $^{33}S$ enrichment ($\Delta^{33}S$ being positive initially) in product $SO_4$; and it is the leftover $SO_2$ being NMD depleted in $^{33}S$ which will later turn into $SO_4$. If true, this "elementary" $SO_2$ to $SO_4$ photo-oxidation step in volcanic plumes would have the opposite sign in $^{33}S$ anomaly to that from tropospheric $SO_2$ oxidation to $SO_4$ as the author proposed. I suggest this difference be discussed.

This is a good point that was worth being mentioned, a point also highlighted by R#2 (see below).

→ A section l.20 p.18 was added:

*" It is worth mentioning that our model would thus generate a different temporal pattern from the one recorded in sulfates from the Antarctica snowpack which are first characterized by positive $\Delta^{33}S$-values that then shift to negative $\Delta^{33}S$-values, reflecting a depletion in $^{33}S$ in the residual $SO_2$ pool (Baroni et al., 2008;Gautier et al., 2018). Although the origin of the $\Delta^{33}S$-values in snowpack remains unclear, a combination of different oxidation pathways with similar contributions of S-MDF (high or lower contribution of OH oxidation pathway) and S-MIF processes (photoexcitation and photolysis) has been recently suggested to explain such $\Delta^{33}S$-values (Gautier et al., 2018). The OH oxidation pathway is occurring in both the troposphere and the stratosphere. However, in the troposphere as i) photolysis cannot occur because of the ozone layer and ii) photooxidation would only occur in a narrow range of UV (see section 4.3.2) but would unlikely display a seasonal variation, we suggest that the reactions responsible for S-MIF in the stratosphere and in the troposphere are different. Thus, the contrasting patterns observed in sulfates in Antarctica and in Montreal could be explained by the implication of different combinations of oxidation pathways where a S-MIF process other than photolysis and photooxidation is involved."*

2. Page 18 line 9-11: Please note that Han et al (2017)'s sulfate were from PM2.5 while Guo et al. (2010) from PM10. The $\Delta33S$ values for Han et al are distinctly negative in winter months while for Guo et al's larger particles are distinctly positive in the months of March to August. Therefore, this pattern is not consistent with the authors' prediction of more negative-$\Delta33S$ sulfate being preferentially found in larger dust particles. I suggest incorporating this difference in your discussion as well.

We agree with the reviewer that including the study of Guo et al. and highlighting the difference in aerosols sizes between Guo et al and Han et al. ($PM_{10}$ versus $PM_{2.5}$) in the discussion is useful. Guo et al.' was not discussed in our original manuscript as the authors did not report chemical analysis for their aerosols, making it impossible to determine whether the $PM_{10}$ could record or be characterized by a higher contribution of dust than $PM_{2.5}$. This is now discussed in the revised version of the manuscript.

→ A section l.33 p.19 was added:

*"Negative $\Delta^{33}S$ values have also not been measured in $PM_{10}$ during spring. Guo et al. (2010) data show positive $\Delta^{33}S$-values, similar to ours and to other studies but different from Han et al. (2017). However, Guo et al. (2010) did not report major*

*elements in their aerosol samples, making it difficult to detect any significant dust contribution. Nevertheless while Guo et al. (2010) measured sulfates S isotope compositions until April 11[th], Cao et al. (2014) reported a significant dust event on April 27[th] of the same year. In that respect this does not contradict our hypothesis: $SO_2$ photooxidation on mineral dust could lead to positive $\Delta^{33}S$ of the residual $SO_2$ transported to Beijing. Moreover, for our model to be consistent with the data of Han et al.*

5     *(2017), their aerosol fine fraction would need to be dominated by dust which is consistent with the observation that Asian dust storms contribute to the $PM_{2.5}$ budget in Beijing (Han et al., 2015)."*

Technical corrections:

Page3 line 3-8: β value in stable isotope community has been reserved for a fundamental concept, i.e. the equilibrium

10     fractionation factor between a compound and its atomic form of element of interest, e.g., the equilibrium fractionation factor between CO2 and O for oxygen isotopes or SO4 and S for sulfur isotopes (Richet et al., 1977). Most in the triple-isotope community use the Greek symbol θ to describe the triple sulfur or triple oxygen isotope relationship, such as 33θ and 17θ., to avoid confusion. If you insist using β, please mention θ.

This is not exactly true: β-factor is indeed used to express the reduced partition function of molecules from which equilibrium

15     fractionation factors are calculated. However, β-values are also used to express mass laws between two isotopic systems in general (e.g. Young and Galy, 2004;) while θ- is specifically used when it corresponds to isotope equilibrium (e.g. Farquhar and Wing, 2003; Dauphas and Schauble, 2016), λ is used to define the slope defined by data which, owing to mass conservation effects, does not necessarily correspond to β- (i.e. the mass exponent relating the two isotope fractionation factors (Farquhar and Wing, 2003, Ono et al., 2006; Johnston et al., 2008). Given the remaining lack of understanding on the reactions involved

20     and the mechanisms (equilibrium, kinetic, etc…), we feel that we cannot use the θ-notation. Using β- is, we think, therefore more appropriate. We have added some clarifications to help the reader with the 'isotope notations' which, we agree, can easily be confusing.

➔ We modified l.5, p3:

" *The β-exponent is usually expressed as θ to refer to isotope equilibrium. We are using β- instead as the processes describing*

25     *the $SO_2$-oxidation are actually not at the isotope equilibrium. Its value depends on the reaction considered (Farquhar et al., 2001;Harris et al., 2013;Ono et al., 2013;Watanabe et al., 2009). At high temperature (> 500°C, i.e. under equilibrium), $^{33}\theta$ and $^{36}\theta$-values are respectively 0.515 and 1.889 (Eldridge et al., 2016;Otake et al., 2008)"*

Page3 Line 7 and 11: If the "deviation" at Line 7 refers only to temperature effect, then Line 11 is ok. Otherwise, Line 11's

30     "Non-zero" cases include the deviation mentioned at Line 7. Therefore, either add the term "temperature" at Line 7 or delete "also" at Line 11.

-➔ This has been modified in the text.

I don't think sulfate with direct or indirect interactions with long-range transported mineral dust would explain the observed $\Delta^{33}S$ patterns, due to the probably small magnitude of this sulfate versus regional or locally produced. The residual $SO_2$ associated with far-away dust source region (from long range transport rooted in Asia or Sahara) maybe just too small to make any difference in $\Delta^{33}S$ measured in Montreal, where local emission of $SO_2$ and the subsequent conversion to sulfate dominate sulfate budget.

This is a good point raised by the reviewer. However it would be possible to account for the $\Delta^{33}S$-values with a small contribution of this oxidation pathway. If 1) the sulfates formed by photooxidation on mineral dust were characterized by high $\Delta^{33}S$-values (hypothetically 10‰), and 2) would hypothetically contribute to ~ 10% of the total sulfate, then even a small contribution of those sulfates, mixed with sulfates formed by the major oxidation pathways, which are locally produced (i.e. $\Delta^{33}S$ ~ 0‰), could explain the $\Delta^{33}S$ -values observed in the troposphere ($\Delta^{33}S < 0.5$).

➔ We have made this point clearer l.15 p.19-:

*"If this latest oxidation pathway could promote the formation of sulfates characterized by high $\Delta^{33}S$-values (hypothetically 10‰), then a small contribution (hypothetically ~10%) from this oxidation pathway would produce a significant signal on the sulfur isotope composition of tropospheric sulfate aerosols (i.e. $\Delta^{33}S$ ~ 1‰ based on these hypotheses). In this case, even a small proportion of those sulfates mixed with sulfates formed by the major oxidation pathways locally produced (i.e. $\Delta^{33}S$ ~ 0‰) could explain the $\Delta^{33}S$-values observed in the troposphere ($\Delta^{33}S <0.5$‰). This hypothesis needs to be further tested."*

In fact, I am confusing by the term of "Photooxidation of SO2 in the presence of mineral dust". as based on the statements in 4.3.4., by Photooxidation of SO2 in the presence of mineral dust, the authors seemed to mean in fact heterogenous SO2 oxidation on the surface of mineral dust or dust enhanced HOx radicals oxidation. If this is the case, then the term of photo-oxidation should be avoided.

Our use of 'photooxidation' is actually meant to be consistent with the literature reporting "photooxidation of $SO_2$ on mineral dust" (Yu et al., 2017;He et al., 2014;Dupart et al., 2012;George et al., 2015;Usher et al., 2003;Zhao et al., 2018;Ma et al., 2018).

what's more, if this is the case, the oxidation then should be no difference from that in gaseous and aqueous phase oxidation in terms of the specific oxidation pathways (or oxidant involved), then why large non-zero D33S could be induced?

We agree about the specific mechanism ($SO_2$ oxidation by OH) but little is known about the number of other mechanisms behind 'the in-particle chemistry'; heterogeneous $SO_2$ oxidation by OH radicals have been described (as mentioned by R#2) but oxidation by other radicals may exist, such as the recently identified superoxide $O_2\cdot$. The S-isotope fractionations associated with this latest oxidation pathway have not been reported yet. Furthermore, others radicals/oxidants might as well be discovered in the future, leading $SO_2$ oxidation by mineral dust to produce specific S-isotope fractionation factors compared to OH-oxidation.

➔ We included R#2's comment and have clarified our point accordingly:

*"To date, the mechanisms behind the in-particle chemistry remain little studied and several $SO_2$ heterogeneous oxidation reactions may have been overlooked"* l.7 p.16.

We also changed the text as follows:- *"The oxidation implicating heterogeneous oxidation and OH radicals should a priori not show significant differences from the one that occurs in gaseous and aqueous phase; i.e. heterogeneous oxidation of $SO_2$ is likely to induce a mass dependent fractionation of S-isotopes (Harris et al., 2012) with resulting negative $\Delta^{33}S$-values <-0.15‰ (Harris et al., 2013). Among other reactions, $SO_2$ oxidation by the $O_2\cdot$ superoxide radical anion is another oxidation reaction that has not yet been isotopically characterized (Dupart et al., 2014;Usher et al., 2003). If this latest oxidation pathway could promote the formation of sulfates characterized by high $\Delta^{33}S$-values (hypothetically 10‰), then a small contribution (hypothetically ~10%) from this oxidation pathway would produce a significant signal on the sulfur isotope composition of tropospheric sulfate aerosols (i.e. $\Delta^{33}S$ ~ 1‰ based on these hypotheses). In this case, even a small proportion of those sulfates mixed with sulfates formed by the major oxidation pathways locally produced (i.e. $\Delta^{33}S$ ~ 0‰) could explain the $\Delta^{33}S$-values observed in the troposphere ($\Delta^{33}S$ <0.5‰). This hypothesis needs to be further tested* " l.10 p.19

If it is really photo-oxidation of SO2 that occurs, I don't understand why photo-oxidation only could occur on the surface of mineral dust. If mineral dust serves as the reaction site promoting the photo-oxidation of SO2, why not other aerosols?

It may well be but, to our knowledge, no study has ever reported photooxidation of $SO_2$ in the presence of other types of C-rich, i.e. 'organic,' aerosols. This could result from the low concentration of metal oxides in other aerosols (Cass et al., 2000). There are studies indicating the photolysis rate of nitrate on aerosols a few orders of magnitude larger than that in the gas phase, could be this the case of SO2 ?

Compared to nitrate, photolysis of the sulfate is less likely because the main wavelength region of $SO_2$-absorption (190-220 nm) is filtered by the ozone layer. However, as stated in section 4.3.2, a narrow wavelength range (typically 320 to 330 nm) where $SO_2$ absorption could occur (i.e. not filtered by the ozone layer) in the troposphere, which still leaves room for S-MIF to be produced. This is better stated in the text l.23 p.15.

What's more, lab experiments, model calculations and ice-core data indicated when photooxidation occurs, the formed sulfate is in general enriched in S-33 and the residual SO2 is depleted in S-33

We have addressed this point above.

Why in this case assuming the opposite pattern? Or just to fulfill the observation?

Indeed, this is inferred, not observed. The text now states more clearly this aspect l.4 p18

1. The different SO2 source in the two cities, especially in winter, heating source should be the main source of SO2 but what is the difference of the energy structure between the two cities?

R#2's comment can be understood in two ways: (a) distinct oxidation processes of sulfur dioxide with $\Delta^{33}S$ ~ 0 ‰ (here low vs high temperature of combustion) leading to markedly distinct isotope signals in Beijing aerosols and (b) distinct sulfur sources having distinct, non-zero $\Delta^{33}S$. Point 'b' is somewhat similar to the comment expressed below by Dr Whitehill.

- a- Both Montréal and Beijing have their energy relying primarily on coal and oil burning. The main difference probably lies in the temperature of combustion of coal and wood (see Han et al., 2016; Lin et al. 2018) where 'low temperature

combustion' would be more significant in Beijing and considered as a possibility to account for the distinctly negative $\Delta^{33}$S of aerosols (Han et al., 2016; Lin et al., 2017). However, Lin et al. (2018) recently questioned this mechanism on the basis of new results (low temperature combustion would not lead to very anomalous $\Delta^{33}$S, yet with still significant non-zero $\Delta^{36}$S-values). We cannot discard this possibility but tried, using available data, to investigate whether SO$_2$ photo-oxidation on mineral dust could represent a viable alternative hypothesis. The revised manuscript states more clearly that additional data are required to discuss such a possibility.

    b-   As suggested by Dr Whitehill in his comment, the question could also relate to distinct sources of sulfur with different $\Delta^{33}$S -values. This is addressed below.

→We modified the discussion l.30 p.12:

-" *This contrasts with the interpretation where the negative $\Delta^{33}$S-values (down to -0.6‰) measured during winter in Beijing would relate to anthropogenic sources, in particular those generating incomplete, i.e. low-temperature, coal or wood combustion (Han et al., 2017). Still, this model cannot explain the total range of isotope compositions observed. The authors mostly rely on data showing that primary aerosols are characterized by negative $\Delta^{33}$S-values but only down to -0.2‰ (Lee et al., 2002). Also the complementary positive $\Delta^{33}$S still need to be addressed. Furthermore, Han et al. (2017) interpretation would predict: i) a seasonality with negative $\Delta^{33}$S-values down to -0.6‰ during winter as a result from increased coal and wood burning and ii) a gradient in the $\Delta^{33}$S-values from the outer towards the inner city with isotope shifting from ~ 0‰ to negative $\Delta^{33}$S-values. This would contradict our observations, since our data in Montreal show the opposite to what was observed in Beijing. It comes that based on the available data of S anthropogenic emissions, the combustion of coal or wood at low temperature can neither explain the $\Delta^{33}$S seasonality nor the highest $\Delta^{33}$S-values up to 0.5‰ measured in urban aerosols.*

*This conclusion is reinforced by the fact that coal is not the major source of energy in Montreal, oil representing 50% of the fuel energy in Quebec (Montréal, 2015). Oil would thus display $\Delta^{33}$S-values close to 0‰ (Lee et al., 2002). Taken together, our observations suggest that anthropogenic activities (both coal and oil combustion) are unlikely responsible for the $\Delta^{33}$S seasonality nor the highest $\Delta^{33}$S-values up to 0.5‰ measured in Montreal urban aerosols. This implies that non-zero $\Delta^{33}$S-values are produced in rural rather than in urban environments. Thus, the following discussion mostly focuses on data from station 98, located on the western part of the island, upstream of the main blowing winds and supposedly less affected by emissions from local anthropogenic activities.*".

2. The aerosols being collected and measured, one is PM2.5 (the fine mode) and the other is PM10 (the coarse mode). The high Na+ concentration in Montreal also indicates the difference, as sea-salt aerosols are often in the coarse mode. Would be sulfate formed in or associated with coarse mode aerosols isotopically different with that in fine mode? It could be another way around, as photo-oxidation of SO2 with coarse mode aerosols leads to sulfate enriched in S33, leaving residual SO2 depleted in S-33 and which is ultimately converted to sulfate by heterogeneous reaction in fine mode aerosols or by gaseous oxidation and then nucleate to or scavenged by fine mode aerosols. Just brainstorming as no concrete answer based on current knowledge available.

The reviewer raises an interesting issue and we agree that S-isotopes of sulfates formed in the aerosol coarse mode could be different from the one in the fine mode fraction. To our knowledge, no study (neither experimental nor with natural samples) has ever been published yet but we fully agree that this is a prediction that can be made from our model, which we have included in the revised manuscript.

Besides, the hypothesis expressed by the reviewer: "It could be another way around,as photo-oxidation of SO2 with coarse mode aerosols leads to sulfate enriched in S33, leaving residual SO2 depleted in S-33 and which is ultimately converted to sulfate by heterogenous reaction in fine mode aerosols or by gaseous oxidation and then nucleate to or scavenged by fine mode aerosols" is also interesting. However, following this hypothesis, positive $\Delta^{33}S$-values would have been explained by the input of sulfates associated with dust, which is not consistent with our chemical analyses that do not indicate any contributions of mineral dust to Montreal aerosols. Moreover, in this case, we would expect negative $\Delta^{33}S$-values on sulfates collected at station 98 (the station least impacted by anthropogenic emissions) which is not the case. For these reasons, this hypothesis cannot account for our observations. We made this clearer in the manuscript.

→ We modified the discussion l.4 p.18 as follows: "*Thus, in order to explain our data (i.e. most positive $\Delta^{33}S$-values at station 98, seasonality of the S-isotope compositions, no dust particles detected in Montreal aerosols), we suggest that the $SO_2$ photooxidation reaction may occur at the dust surface and, by oxidizing the surrounding $SO_2$ into sulfates, it would deplete the resulting $SO_4$ in $^{33}S$ and by mass balance, leave the residual $SO_2$ enriched in $^{33}S$ (Figure 6). Sulfates associated to dust would be characterized by negative $\Delta^{33}S$-values and will be deposited while the residual atmospheric $SO_2$ (i.e. characterized by positive $\Delta^{33}S$-values) would be transported to Montreal. The transported $SO_2$ enriched in $^{33}S$ would then be oxidized into sulfates in Montreal vicinity through the major oxidation pathways ($O_2$+TMI, $H_2O_2$, $O_3$, OH, $NO_2$). We suggest the presence of two different types of sulfates: i) the first type would be formed by photooxidation and would be associated to coarse particles (dust particles) while ii) the second type would be formed by the oxidation of the remaining $SO_2$ and thus likely be associated to finer particles. These sulfates supposedly characterized by positive $\Delta^{33}S$ up to 0.5‰ would be mixed with both primary and secondary sulfates emitted and formed within the city and supposedly characterized by $\Delta^{33}S$-values close to 0‰ (i.e. oxidation by $O_2$+TMI, $H_2O_2$, $O_3$, OH, $NO_2$ ; Figure 6).* "

"

**Comment #1 : Dr. Andrew Whitehill**

I would like to see more discussion about the $\Delta^{33}S$ values of the initial $SO_2$ and more justifications / measurements / citations for these assumptions. Iron production in the Quebec / Ontario region of Canada is a high emitter of sulfur dioxide. My first order assumption would be that processing iron from Archean banded iron formations would release $SO_2$ with non mass dependent isotope signatures. $SO_2$ signatures from this region can be transported long distances and may contribute to non mass dependent isotope signatures significantly downwind (e.g. Boston, MA). It would be useful to understand your reasoning as to why either (1) $SO_2$ emitted from processing iron from banded iron formations will not produce non mass dependent $SO_2$ or (2) this is not a substantial source of $SO_2$ for this region and will not affect the observed isotope signatures. You invoke

complex reactions (e.g. $SO_2$ photooxidation and stabilized Criegee intermediates) without constraining the $SO_2$ source signature. A mixing of $SO_2$ from different sources would have different $\delta^{34}S$ (and likely $\Delta^{33}S$ and $\Delta^{36}S$) values and may contain seasonality as observed in this study.

Below are the answers to the two questions.

5    It would be useful to understand your reasoning as to why (2) this is not a substantial source of $SO_2$ for this region and will not affect the observed isotope signatures

The large majority of coal and oil used worldwide for energy are derived from Proterozoic sediments (<2.3 Gy) and, as such, does not have significant non-zero $\Delta^{33}S$ (typically within ±0.1‰, e.g. *Farquhar and Wing, 2003*). The complete conversion of sulfur (as organic S, sulfate and/or pyrite) to $SO_2$ implies that it would have the same isotope composition than that of its

10    starting material, i.e. no isotope fraction and $\Delta^{33}S$ = 0.0±0.1‰. Only if part of the $SO_2$ is scavenged and the fractionation process is strongly non-mass dependent (beta $\neq$ 0.515) would the emitted $SO_2$ have non-zero $\Delta^{33}S$. 'Low temperature combustion' was suggested to represent such a process. However, this is dealing with identifying processes. Dr Whithill and R#2 wonders whether the source of sulfur could be characterized by non-zero $\Delta^{33}S$, with an emphasis on iron mining, which relies on the extraction of some Archean banded-iron formation mining. These kinds of samples can have high S-content

15    (several percent) with significant non-zero $\Delta^{33}S$ (typically positive $\Delta^{33}S$-values, up to several per mille). This is a possibility that we did not originally address.

Iron production in the Quebec / Ontario region of Canada is indeed an emitter of sulfur dioxide. However, with a total of 5800 tons of $SO_2$ emission each year (Environnement Canada, 2013), this only represents 1.5% of the total 370000 tons of $SO_2$ emitted (Environnement Canada, 2013).

20    It would be useful to understand your reasoning as to why (1) $SO_2$ emitted from processing iron from banded iron formations will not produce non mass dependent $SO_2$

By considering that 1) the BIF are characterized by a $\Delta^{33}S$-values up to 2‰ (Thomassot et al., 2015), 2) iron process does not fractionate the sulfur isotopes (similar to coal combustion) and 3) only 1.5% of the $SO_2$ results from iron processing (Environnement Canada, 2013), this would only produce $\Delta^{33}S$-anomalies up to 0.02‰. Thus, iron processing can hardy

25    account for the origin of non-zero $\Delta^{33}S$-values observed in most aerosols.

More specifically, Canadian iron ore production is split in ~50-45% between Quebec and Labrador (with 5% in Nunavut). With respect to Quebec, iron production is mainly from Algoma BIFs of about 2.8-2.7 Ga, typified by the Temagami deposit for which there are recently available $^{33}S$ data (Diekrup et al., 2018). From their Table 1, all their sedimentary data (oxidic facies, cherts, BIF sulphides; sulphidic clays; sulphide veins) gives an average of 0.467 +/- 0.707 (one standard deviation;

30    n=50), which makes emitted $SO_2$ having even smaller $\Delta^{33}S$.

→ Section 4.1 " Anthropogenic emission, $\Delta^{33}S$ -values and seasonality" has been modified "l.3 p.12 : *"The large majority of coal and oil used worldwide as an energy source are extracted from Proterozoic sediments (<2.3 Gy) and, as such, does not have significant non-zero $\Delta^{33}S$ (typically within ±0.1‰, e.g. Farquhar and Wing, 2003). The complete conversion of sulfur (as organic S, sulfate and/or pyrite) to $SO_2$ implies that $SO_2$ has the same isotope composition than that of its starting material,*

*i.e. no isotope fractionation or $\Delta^{33}S = 0.0\pm0.1$‰. Only if part of the $SO_2$ is scavenged and the fractionation process is strongly non-mass dependent ($\beta \neq 0.515$) would the emitted $SO_2$ have a non-zero $\Delta^{33}S$. Han et al. (2016) suggested that the combustion of coal or wood at low temperature may represent such conditions.*

*Iron extraction from Archean banded-iron formation (BIF) is another source of atmospheric S that therefore produces non-zero $\Delta^{33}S$. However, the 5,800 tons of $SO_2$ emitted each year in the Quebec/Ontario region by mining activities (Environnement Canada, 2013) only represents 1.5% of the annual 370,000 tons of national $SO_2$ emissions (Environnement Canada, 2013). If we consider a high average $\Delta^{33}S$ of 2‰ (Thomassot et al., 2015) and a proportion of 1.5% of $SO_2$ resulting from iron processing, this would lead to an average $\Delta^{33}S$-anomaly of the final $SO_2$ of up to 0.02‰. Thus, iron processing can hardy account for the origin of the non-zero $\Delta^{33}S$-values observed in most aerosols. More specifically, the Canadian iron ore production is split between Quebec (50%), Labrador (45%) and Nunavut (5%). With respect to Quebec, iron production is mainly operated from the Algoma BIFs (~2.8 Ga) typified by the Temagami deposits for which Diekrup et al. (2018) give an average $\Delta^{33}S$ of $0.467\pm0.707$ (samples including oxidic facies, cherts, BIF sulphides and sulphidic clays, sulphide veins), which makes the emitted $SO_2$ having even smaller $\Delta^{33}S$. In the following discussion we will therefore consider that sources of $SO_2$ have $\Delta^{33}S$~0‰ and that only specific chemical reactions (photochemical or not) can produce non-zero $\Delta^{33}S$.*"

[revised manuscript text omitted]

$$^{3y}\alpha = (^{34}\alpha)^{3y\beta} \qquad (3)$$

where $^{3y}\alpha$ is either $^{33}\alpha$ or $^{36}\alpha$ and $^{3y}\beta$ is either $^{33}\beta$ or $^{36}\beta$. The $^{3y}\beta$-exponent describes the relative fractionation of $^{3y}S/^{32}S$ and $^{34}S/^{32}S$. The $\beta$-exponent is usually expressed as $\theta$ to refer to isotope equilibrium. We are using $\beta$- instead as the processes describing the SO$_2$-oxidation are actually not at the isotope equilibrium. Its value depends on the reaction considered (Farquhar et al., 2001;Harris et al., 2013a;Ono et al., 2013;Watanabe et al., 2009). At high temperature (> 500°C, i.e. under equilibrium), $^{33}\theta$ $\beta$ and $^{36}\theta$ $\beta$-values are respectively 0.515 and 1.889 (Eldridge et al., 2016;Otake et al., 2008). Deviation of the $^{3y}\beta$-value from these high temperature values usually leads to non-zero $\Delta^{33}S$ and $\Delta^{36}S$ values typically in the range of ±0.1‰ and ±1‰, respectively. $\Delta^{33}S$ and $\Delta^{36}S$ are expressed as follows (Farquhar and Wing (2003)):

$$\Delta^{33}S = (\delta^{33}S + 1) - (\delta^{34}S + 1)^{0.515} \qquad (4)$$

$$\Delta^{36}S = (\delta^{36}S + 1) - (\delta^{34}S + 1)^{1.889} \qquad (5)$$

[revised manuscript text omitted]

**4    Discussion**

**4.1    Anthropogenic emissions, $\Delta^{33}$S-values and seasonality**

The large majority of coal and oil used worldwide as an energy source are extracted from Proterozoic sediments (<2.3 Gy) and, as such, does not have significant non-zero $\Delta^{33}$S (typically within ±0.1‰, e.g. Farquhar and Wing, 2003). The complete conversion of sulfur (as organic S, sulfate and/or pyrite) to $SO_2$ implies that $SO_2$ has the same isotope composition than that of its starting material, i.e. no isotope fractionation or $\Delta^{33}$S = 0.0±0.1‰. Only if part of the $SO_2$ is scavenged and the fractionation process is strongly non-mass dependent (ß ≠ 0.515) would the emitted $SO_2$ have a non-zero $\Delta^{33}$S. Iron extraction from Archean banded-iron formation (BIF) is another source of atmospheric S that therefore produces non-zero $\Delta^{33}$S. However, the 5,800 tons of $SO_2$ emitted each year in the Quebec/Ontario region by mining activities (Environnement Canada, 2013) only represents 1.5% of the annual 370,000 tons of national $SO_2$ emissions (Environnement Canada, 2013). If we consider a high average $\Delta^{33}$S of 2‰ (Thomassot et al., 2015) and a proportion of 1.5% of $SO_2$ resulting from iron processing, this would lead to an average $\Delta^{33}$S-anomaly of the final $SO_2$ of up to 0.02‰. Thus, iron processing can hardy account for the origin of the non-zero $\Delta^{33}$S-values observed in most aerosols. More specifically, the Canadian iron ore production is split between Quebec (50%), Labrador (45%) and Nunavut (5%). With respect to Quebec, iron production is mainly operated from the Algoma BIFs (~2.8 Ga) typified by the Temagami deposits for which Diekrup et al. (2018) give an average $\Delta^{33}$S of 0.467±0.707‰ (samples including oxidic facies, cherts, BIF sulphides and sulphidic clays, sulphide veins), which makes the emitted $SO_2$ having even smaller $\Delta^{33}$S. In the following discussion we will therefore consider that sources of $SO_2$ have $\Delta^{33}$S~0‰ and that only specific chemical reactions (photochemical or not) can produce non-zero $\Delta^{33}$S.

Aerosols collected at stations likely impacted by local emission sources (i.e. stations 03, 06, 13 and 50) present the lowest $\Delta^{33}$S-values (~ -0.01‰) and the highest $\delta^{34}$S-values (up to ~ 12‰) compared to station 98 (less influenced by anthropogenic emissions). This suggests that local emissions in Montreal are characterized by $\delta^{34}$S-values up to 12‰ and mean $\Delta^{33}$S-values close to 0‰, which implies that the high $\Delta^{33}$S-anomalies with lower $\delta^{34}$S are transported to rather than produced in Montreal. Local anthropogenic sources could then isotopically impact these imported aerosol sulfates by decreasing their $\Delta^{33}$S-values towards 0‰. This is consistent with Lee et al. (2002) who showed the ability of these primary aerosols, resulting from the combustion process, to decrease $\Delta^{33}$S-values towards 0‰, as they are characterized by zero to slightly negative $\Delta^{33}$S-values down to -0.2‰. Secondary sulfates formed by $SO_2$ oxidation within cities by the main oxidation pathways (OH, $O_2$ + TMI, $NO_2$, $H_2O_2$, $O_3$) would not generate significant MIF and would also lead to decrease the $\Delta^{33}$S-value of imported aerosols. (Harris et al., 2013a; Au Yang et al., 2018).

This contrasts with the interpretation where the negative $\Delta^{33}$S-values (down to -0.6‰) measured during winter in Beijing would relate to anthropogenic sources, in particular those generating incomplete , i.e. low-temperature, coal or wood combustion (Han et al., 2017). Still, this model cannot explain the total range of isotope compositions observed. T-he authors

mostly rely on data showing that primary aerosols are characterized by negative $\Delta^{33}$S-values but only down to -0.2‰ (Lee et al., 2002). Although the complementary positive $\Delta^{33}$S still need to be addressed. Furthermore, (Han et al., 2017) interpretation would predict: i) a seasonality with negative $\Delta^{33}$S-values down to -0.6‰ during winter as a result from increased coal and wood burning and ii) a gradient in the $\Delta^{33}$S-values from the outer towards the inner city with isotope shifting from ~ 0‰ to negative $\Delta^{33}$S-values. This would contradict our observations, since our data in Montreal show the opposite to what was observed in Beijing. It comes that based on the available data of S anthropogenic emissions, the combustion of coal or wood at low temperature can neither explain the $\Delta^{33}$S seasonality nor the highest $\Delta^{33}$S-values up to 0.5‰ measured in urban aerosols.

This conclusion is reinforced by the fact that coal is not the major source of energy in Montreal, oil representing 50% of the fuel energy in Quebec (Ville de Montréal, 2015). Oil would thus display $\Delta^{33}$S-values close to 0‰ (Lee et al., 2002). Taken together, our observations suggest that anthropogenic activities (both coal and oil combustion) are unlikely responsible for the $\Delta^{33}$S seasonality nor the highest $\Delta^{33}$S-values up to 0.5‰ measured in Montreal urban aerosols. This implies that non-zero $\Delta^{33}$S-values are produced in rural rather than in urban environments. Thus, the following discussion mostly focuses on data from station 98, located on the western part of the island, upstream of the main blowing winds and supposedly 
[revised manuscript text omitted]

10 pathways ($O_2$+TMI, $H_2O_2$, $O_3$, OH, $NO_2$). We suggest the presence of two different types of sulfates: i) the first type would be formed by photooxidation and would be associated to coarse particles (dust particles) while ii) the second type would be formed by the oxidation of the remaining $SO_2$ and thus likely be associated to finer particles,. These sulfates supposedly characterized by positive $\Delta^{33}$S up to 0.5‰ would be mixed with both primary and secondary sulfates emitted and formed within the city and supposedly characterized by $\Delta^{33}$S-values close to 0‰ (i.e. oxidation by $O_2$+TMI, $H_2O_2$, $O_3$, OH, $NO_2$ ;

15 Figure 6

[Figure]

[Figure]

**Figure 6 : Scheme of the reaction mechanisms leading to the formation of sulfates  driven by airborne mineral dust. The $^{33}$S enriched residual SO$_2$ is supposedly transported to cities while the $^{33}$S depleted sulfates is deposited along with dust particles in the rural environment. .**

It is worth mentioning that our model would thus generate a different temporal pattern from the one recorded in sulfates from the Antarctica snowpack which are first characterized by positive $\Delta^{33}$S-values that then shift to negative $\Delta^{33}$S-values, reflecting a depletion in $^{33}$S in the residual $SO_2$ pool (Baroni et al., 2008;Gautier et al., 2018). Although the origin of the $\Delta^{33}$S-values in snowpack remains unclear, a combination of different oxidation pathways with similar contributions of S-MDF (high or lower contribution of OH oxidation pathway) and S-MIF processes (photoexcitation and photolysis) has been recently suggested to explain such $\Delta^{33}$S-values (Gautier et al., 2018). The OH oxidation pathway is occurring in both the troposphere and the stratosphere. However, in the troposphere as i) photolysis cannot occur because of the ozone layer and ii) photooxidation would only occur in a narrow range of UV (see section 4.3.2) but would unlikely display a seasonal variation, we suggest that the reactions responsible for S-MIF in the stratosphere and in the troposphere are different. Thus, the contrasting patterns observed in sulfates in Antarctica and in Montreal could be explained by the implication of different combinations of oxidation pathways where a S-MIF process other than photolysis and photooxidation is involved.

It is worth noting that the exact photochemical mechanism which would be responsible for that relation remains speculative but some reactions can be highlighted and discussed. The recently proposed oxidation of $SO_2$ by $NO_2$ on mineral dust (Ma et al., 2018) is unlikely because $NO_2$ is predominant in the urban environment, i.e. at odds with the present evidence. The oxidation implicating heterogeneous oxidation and OH radicals should *a priori* not show significant differences from the one that occurs in gaseous and aqueous phase; i.e. heterogeneous oxidation of $SO_2$ is likely to induce a mass dependent fractionation of S-isotopes (Harris et al., 2012a) while the gas phase by OH would induce negative $\Delta^{33}$S-values $\leq$ -0.15‰ (Harris et al., 2013a). Among other reactions, $SO_2$ oxidation by the $O_2\cdot$ superoxide radical anion is another oxidation reaction that has not yet been isotopically characterized (Dupart et al., 2014;Usher et al., 2003). If this latest oxidation pathway could promote the formation of sulfates characterized by high $\Delta^{33}$S-values (hypothetically 10‰), then a small contribution (hypothetically ~10%) from this oxidation pathway would produce a significant signal on the sulfur isotope composition of tropospheric sulfate aerosols (i.e. $\Delta^{33}$S ~ 1‰ based on these hypotheses). In this case, even a small proportion of those sulfates mixed with sulfates formed by the major oxidation pathways locally produced (i.e. $\Delta^{33}$S ~ 0‰) could explain the $\Delta^{33}$S-values observed in the troposphere ($\Delta^{33}$S <0.5‰). This hypothesis needs to be further tested in the future.

Our hypothesis could explain sulfates with positive $\Delta^{33}$S-values transported to Montreal but implies that negative $\Delta^{33}$S-values also need to be found in dust particles. This hypothesis could leave a new room to explain negative $\Delta^{33}$S-values measured in Beijing aerosols (Han et al., 2017). Indeed this oxidation pathway may occur at a larger scale and may also be involved in the formation of urban aerosols reported in the literature. Indeed, urban aerosol sulfates previously studied in La Jolla, Bakersfield and White Mountain in the United States, and in Xianghe and Beijing may also be influenced by Asian mineral dust.

Intuitively, dust particles may be transported during discrete storm episodes (Marticorena and Bergametti, 1995;Kok et al., 2012) which have been reported mostly during spring in China (Zhao et al., 2006;An et al., 2018). Following this hypothesis, negative $\Delta^{33}$S-values would be found in spring, which is not the case in both $PM_{2.5}$ and $PM_{10}$ collected in 2016 and 2005, respectively (Guo et al., 2010;Han et al., 2017). In fact, five dust episodes were identified in China in 2016 (An et al., 2018) with one (March 4[th]) happening close to the sampling period (March 15[th] to April 26[th]; Han et al. (2017)). However, the images recorded by the NASA satellite show that the dust storm in the Gobi Desert would unlikely reach Beijing that day (https://modis.gsfc.nasa.gov/gallery/individual.php?db_date=2016-03-11), possibly explaining why such negative values have not been measured Han et al. (2017). Negative $\Delta^{33}$S values have also not been measured in $PM_{10}$ during spring. Guo et al. (2010) data show positive $\Delta^{33}$S-values, similar to ours and to other studies but different from Han et al. (2017). However, Guo et al. (2010) did not report major elements in their aerosol samples, making it difficult to detect any significant dust contribution. Nevertheless while Guo et al. (2010) measured sulfates S isotope compositions until April 11[th], Cao et al. (2014) reported a significant dust event on April 27[th] of the same year. In that respect this does not contradict our hypothesis : $SO_2$ photooxidation on mineral dust could lead to positive $\Delta^{33}$S of the residual $SO_2$ transported to Beijing. Moreover, for our model to be consistent with the data of Han et al. (2017), their aerosol fine fraction would need to be dominated by dust which is consistent with the observation that Asian dust storms contribute to the $PM_{2.5}$ budget in Beijing (Han et al., 2015).

[revised manuscript text omitted]

50  Ville de Montréal: Reduced dependance on fossil fuels in Montréal 2015.

Wadleigh, M., Schwarcz, H., and Kramer, J.: Isotopic evidence for the origin of sulphate in coastal rain, Tellus B, 48, 44-59, 1996.

Wagener, T., Guieu, C., Losno, R., Bonnet, S., and Mahowald, N.: Revisiting atmospheric dust export to the Southern Hemisphere ocean: Biogeochemical implications, Global Biogeochemical Cycles, 22, 2008.

Wall, S. M., John, W., and Ondo, J. L.: Measurement of aerosol size distributions for nitrate and major ionic species, Atmospheric 55 Environment (1967), 22, 1649-1656, 1988.

Wasiuta, V., Lafrenière, M. J., Norman, A.-L., and Hastings, M. G.: Summer deposition of sulfate and reactive nitrogen to two alpine valleys in the Canadian Rocky Mountains, Atmospheric Environment, 101, 270-285, 2015.

Watanabe, Y., Farquhar, J., and Ohmoto, H.: Anomalous fractionations of sulfur isotopes during thermochemical sulfate reduction, Science, 324, 370-373, 2009.

Whitehill, A., Jiang, B., Guo, H., and Ono, S.: SO 2 photolysis as a source for sulfur mass-independent isotope signatures in stratosphehric aerosols, Atmospheric Chemistry and Physics, 15, 1843-1864, 2015.

Whitehill, A. R., and Ono, S.: Excitation band dependence of sulfur isotope mass-independent fractionation during photochemistry of sulfur dioxide using broadband light sources, Geochimica et Cosmochimica Acta, 94, 238-253, 2012.

Whitehill, A. R., Xie, C., Hu, X., Xie, D., Guo, H., and Ono, S.: Vibronic origin of sulfur mass-independent isotope effect in photoexcitation of SO2 and the implications to the early earth's atmosphere, Proceedings of the National Academy of Sciences, 110, 17697-17702, 2013.

Ambient (outdoor) air quality and health: http://www.who.int/mediacentre/factsheets/fs313/en/, access: 11/29/17, 2016.

Yang, Y., Russell, L. M., Lou, S., Liao, H., Guo, J., Liu, Y., Singh, B., and Ghan, S. J.: Dust-wind interactions can intensify aerosol pollution over eastern China, Nature communications, 8, 15333, 2017.

Yang, Y., Wang, H., Smith, S. J., Zhang, R., Lou, S., Qian, Y., Ma, P.-L., and Rasch, P. J.: Recent intensification of winter haze in China linked to foreign emissions and meteorology, Scientific reports, 8, 2107, 2018.

Ye, J., Abbatt, J. P., and Chan, A. W.: Novel pathway of SO 2 oxidation in the atmosphere: reactions with monoterpene ozonolysis intermediates and secondary organic aerosol, Atmospheric Chemistry and Physics, 18, 5549-5565, 2018.

Young, E. D., Galy, A., and Nagahara, H.: Kinetic and equilibrium mass-dependent isotope fractionation laws in nature and their geochemical and cosmochemical significance, Geochimica et Cosmochimica Acta, 66, 1095-1104, 2002.

Yu, F., and Luo, G.: Simulation of particle size distribution with a global aerosol model: contribution of nucleation to aerosol and CCN number concentrations, Atmospheric Chemistry and Physics, 9, 7691-7710, 2009.

Yu, Z., Jang, M., and Park, J.: Modeling atmospheric mineral aerosol chemistry to predict heterogeneous photooxidation of SO 2, Atmospheric Chemistry and Physics, 17, 10001-10017, 2017.

Zhao, D., Song, X., Zhu, T., Zhang, Z., Liu, Y., and Shang, J.: Multiphase oxidation of SO 2 by NO 2 on CaCO 3 particles, Atmospheric Chemistry and Physics, 18, 2481-2493, 2018.

Zhao, T., Gong, S., Zhang, X., Blanchet, J.-P., McKendry, I., and Zhou, Z.: A simulated climatology of Asian dust aerosol and its trans-Pacific transport. Part I: Mean climate and validation, Journal of Climate, 19, 88-103, 2006.

Zinger, I., and Delisle, C.: Quality of used-snow discharged in the ST-Lawrence river, in the region of the Montreal Harbor, Water, Air, and Soil Pollution, 39, 47-57, 1988.